# Sphingosine d18:1 promotes nonalcoholic steatohepatitis by inhibiting macrophage HIF-2α

Jialin Xia[1,2,3,4,5,11], Hong Chen[1,2,3,4,5,11], Xiaoxiao Wang[6,11], Weixuan Chen[4,5], Jun Lin[1,2,3], Feng Xu[1,2,3], Qixing Nie[1,2,3], Chuan Ye [1,2,3], Bitao Zhong[1], Min Zhao [4,5], Chuyu Yun[4,5], Guangyi Zeng[1,2,3,4,5], Yuejian Mao[5], Yongping Wen[7], Xuguang Zhang[7,8], Sen Yan [4,5], Xuemei Wang[1,2,3], Lulu Sun[5,9], Feng Liu[6], Chao Zhong [10], Pengyan Xia [10], Changtao Jiang [1,2,3,10], Huiying Rao [6] ✉ & Yanli Pang [1,4,5] ✉

Non-alcoholic steatohepatitis (NASH) is a severe type of the non-alcoholic fatty liver disease (NAFLD). NASH is a growing global health concern due to its increasing morbidity, lack of well-defined biomarkers and lack of clinically effective treatments. Using metabolomic analysis, the most significantly changed active lipid sphingosine d18:1 [So(d18:1)] is selected from NASH patients. So(d18:1) inhibits macrophage HIF-2α as a direct inhibitor and promotes the inflammatory factors secretion. Male macrophage-specific HIF-2α knockout and overexpression mice verified the protective effect of HIF-2α on NASH progression. Importantly, the HIF-2α stabilizer FG-4592 alleviates liver inflammation and fibrosis in NASH, which indicated that macrophage HIF-2α is a potential drug target for NASH treatment. Overall, this study confirms that So(d18:1) promotes NASH and clarifies that So(d18:1) inhibits the transcriptional activity of HIF-2α in liver macrophages by suppressing the interaction of HIF-2α with ARNT, suggesting that macrophage HIF-2α may be a potential target for the treatment of NASH.

With lifestyle changes, nonalcoholic fatty liver disease (NAFLD) has become a major chronic disease in contemporary society[1]. NAFLD is a chronic metabolic disease characterized by excessive accumulation of fat in hepatocytes. NAFLD can be divided into simple fatty liver (NAFL) and nonalcoholic steatohepatitis (NASH)[2]. Chronic liver injury in NASH significantly increases the risk of end-stage liver diseases (such as cirrhosis and liver cancer). However, there is no effective drug for NASH in clinical practice[3]. Therefore, clarifying the key molecular mechanism

[1]Department of Physiology and Pathophysiology, School of Basic Medical Sciences, Center for Reproductive Medicine, Third Hospital, Peking University, Beijing, China. [2]Center of Basic Medical Research, Institute of Medical Innovation and Research, Peking University Third Hospital, Beijing, China. [3]Center for Obesity and Metabolic Disease Research, School of Basic Medical Sciences, Peking University, Beijing, China. [4]State Key Laboratory of Female Fertility Promotion, Center for Reproductive Medicine, Department of Obstetrics and Gynecology, Peking University Third Hospital, Beijing, China. [5]Institute of Advanced Clinical Medicine, Peking University, Beijing, China. [6]Peking University People's Hospital, Peking University Hepatology Institute, Beijing Key Laboratory of Hepatitis C and Immunotherapy for Liver Diseases, Beijing International Cooperation Base for Science and Technology on NAFLD Diagnosis, Beijing, China. [7]Mengniu Institute of Nutrition Science, Shanghai, China. [8]Shanghai Institute of Nutrition and Health, The Chinese Academy of Sciences, Shanghai, China. [9]Department of Endocrinology and Metabolism, Peking University Third Hospital, Beijing, China. [10]Department of Immunology, School of Basic Medical Sciences, NHC Key Laboratory of Medical Immunology, Medicine Innovation Center for Fundamental Research on Major Immunology-related Diseases, Peking University, Beijing, China. [11]These authors contributed equally: Jialin Xia, Hong Chen, Xiaoxiao Wang. ✉e-mail: raohuiying@pkuph.edu.cn; yanlipang@bjmu.edu.cn

of the occurrence and development of NASH will contribute to developing new strategies for anti-NASH treatment.

The classical theory of NASH pathogenesis is that NASH is caused by the excessive accumulation of lipids in hepatocytes. Then, extreme oxidative stress and inflammation further induce hepatocyte death and the development of inflammation and fibrosis[4]. Sphingolipids are lipids with high biological activity and are one of the main factors affecting the progression of NASH. Sphingolipids mainly include ceramide, sphingosine-1-phosphate (S1P), sphingomyelins, and sphingosine[5]. Previous studies have found changes in ceramide and S1P levels in NASH patients[6]. However, they both failed to act as sensitive biomarkers to guide disease diagnosis in NASH because of their widespread variation in many early-stage NAFLD patients. Sphingosine has not only been found to vary in NAFL[7,8] but has even been found to be useful as a biomarker to predict cirrhosis[9].

Here, we found that So(d18:1) increases significantly in patients with NASH by metabolomics profiling analysis. So(d18:1) promotes liver inflammation and fibrosis in the NASH model. RNA-seq data revealed that So(d18:1) inhibits HIF-2α expression. Macrophage-specific knockout or overexpression of HIF-2α has been used to clarify the role of macrophage HIF-2α in NASH development. Mechanistically, So(d18:1) inhibits macrophage HIF-2α by inhibiting its combination with ARNT and then promotes the secretion of inflammatory factors. Notably, we found that the pharmacological activation of macrophage HIF-2α by FG-4592, a HIF prolyl hydroxylase inhibitor that is approved for the treatment of anaemia in China, had preventive effects on NASH in mice. This work suggests that macrophage HIF-2α is a potential target for the treatment of NASH.

## Results

### Disturbances in sphingolipid metabolism in NASH patients

In the Chinese patient population, we employed a metabolomics screen of NASH patients and healthy volunteers (Supplementary Table 1). The results showed that the changes in the sphingolipid pathway are the most concentrated, significant and dramatic compared to other lipids that are considered to change routinely (Fig. 1A). After that we further examined the whole sphingolipidome using targeted metabolomics (Fig. 1B). Principal component analysis (PCA) showed a clear separation between the healthy volunteers and NASH patients (Fig. 1C). The VIP score indicated a significant increase in the levels of several sphingolipids, especially So(d18:1) (Fig. 1D).

In the human cohort, So(d18:1) accumulated largely in the serum of NASH patients (Fig. 1E, F). Moreover, the concentration of So(d18:1) was positively correlated with serum ALT, AST levels and Fibrosacn index (Fig. 1H–J). To further validate this phenotype, we also fed the mice with CDAA-HFD for 8 weeks to establish the NASH mice model. Serum So(d18:1) concentrations were assayed in NASH model mice, and the trend of increasing serum So(d18:1) concentrations in mice was exactly the same as the trend of increasing ALT and AST levels (Fig. 1G, Fig. S1A, B). These results suggested that So(d18:1) concentrations may be closely related to NASH progression. However, So(d18:1) relative concentration in whole liver tissue didn't increase with the development of NASH (Fig. S1C). The origin of So(d18:1) may need further investigation.

In our sphingolipidome results, the upstream and downstream metabolites of sphingosine, ceramide and S1P, were also altered in content. In our previous study, we found that ceramide was enriched in NASH patients similarly[6]. But ceramides did not increase more with disease progression (Fig. S1D). S1P and the other types of sphingosines also didn't show any growth trends during the progression of NASH (Fig. S1E–I). These results further demonstrated the unique indicative role of So(d18:1) in the progression of NASH.

Hepatic steatosis and lobular inflammation are two important features of NASH. We analysed the changes of So(d18:1) levels in these two aspects. There was no significant increase in the So(d18:1) level as

hepatic steatosis progressed (Fig. S1J). However, the So(d18:1) concentration shows a gradual trend of increase with the aggravation of lobular inflammation (Fig. S1K), which suggests that the function of So(d18:1) may be related to lobular inflammation.

### So(d18:1) aggravates inflammation and fibrosis in NASH

To test whether So(d18:1) is involved in the progression of NASH, CDAA-HFD-fed mice were intraperitoneally injected with vehicle, So(d16:1) or So(d18:1), respectively. So(d16:1) was used as a control to show the specific effects of So(d18:1). Detection of the concentration of So(d18:1) in the plasma after injection shows that the growth of So(d18:1) is similar to the concentration in the plasma of NASH patients, which can mimic the function of So(d18:1) in the pathological state (Fig. S2A).

Liver weight and the ratio of liver weight to body weight were significantly increased in mice injected with So(d18:1) compared with mice injected with vehicle (Figs. 2A, B and S2B). The levels of ALT and AST in the serum of mice injected with So(d18:1) were significantly higher than those in control mice (Fig. 2C, D), which suggests that So(d18:1) exacerbated liver damage in mice. While there were no differences in liver triglyceride (TG), serum TG and serum non-esterified fatty acid (NEFA) levels, there was also no difference in liver and serum cholesterol (CE) levels (Fig. S2C–G). For a clearer image of the liver damage in mice, we made pathological sections and performed H&E staining and Sirius red staining. The pathological sections showed that So(d18:1) treatment increased the fibrosis, lobular inflammation and the histology score of NAFLD activity but did not affect the histology score of hepatic steatosis (Fig. 2E–J). Consistently, the mRNA expression of inflammation genes and fibrosis genes were significantly upregulated in the liver of the So(d18:1) group compared with that of the vehicle group (Fig. 2K, L), while the lipid metabolism genes were mostly not different (Fig. S2H). The results showed that the administration of So(d16:1) to mice of the NASH model did not further exacerbate the disease phenotype of NASH (Figs. 2 and S2). This reinforced the significant role of So(d18:1). Collectively, these results suggest that So(d18:1) can exacerbate lobular inflammation and fibrosis in the livers of NASH mice.

### So(d18:1) inhibits HIF-2α transcription function in liver macrophages

So(d18:1) can exacerbate lobular inflammation in the liver of NASH, suggesting that it mainly alters the immune status of the liver, so we focused on immune cells for an in-depth study. To confirm the changes of various immune cells during the development of NASH, a set of public single-cell RNA-sequencing data from the livers of NASH mice were located and analysed[10]. The results showed an increase in all kinds of immune cells in the livers of NASH mice. However, the largest proportion of these cells were macrophages and monocytes. Importantly, they were recruited to the livers much earlier than other immune cells (Fig. S3A, B). We therefore wanted to see whether So(d18:1) would also cause changes in macrophage proportion. We administered So(d18:1) or So(d16:1) intraperitoneally to chow-diet fed mice for 1 week. Results showed that only So(d18:1) increased the proportion of liver macrophages among all immune cells (Figs. S3C and 3A, B).

To search for the mechanisms by which So(d18:1) promotes macrophage activation, we treated mouse bone marrow-derived macrophages (BMDM) with So(d18:1) or control vehicle under inflammatory stimulation and performed RNA sequencing to explore the changed gene pathways. GO:BP pathway enrichment showed that hypoxia-related pathways were changed significantly between the control and So(d18:1) groups (Fig. 3C). There are two transcription factors that play a major role in the hypoxia-related signalling pathway, HIF-1α and HIF-2α. We further targeted the signalling pathways regulated by these two transcription factors for enrichment analysis.

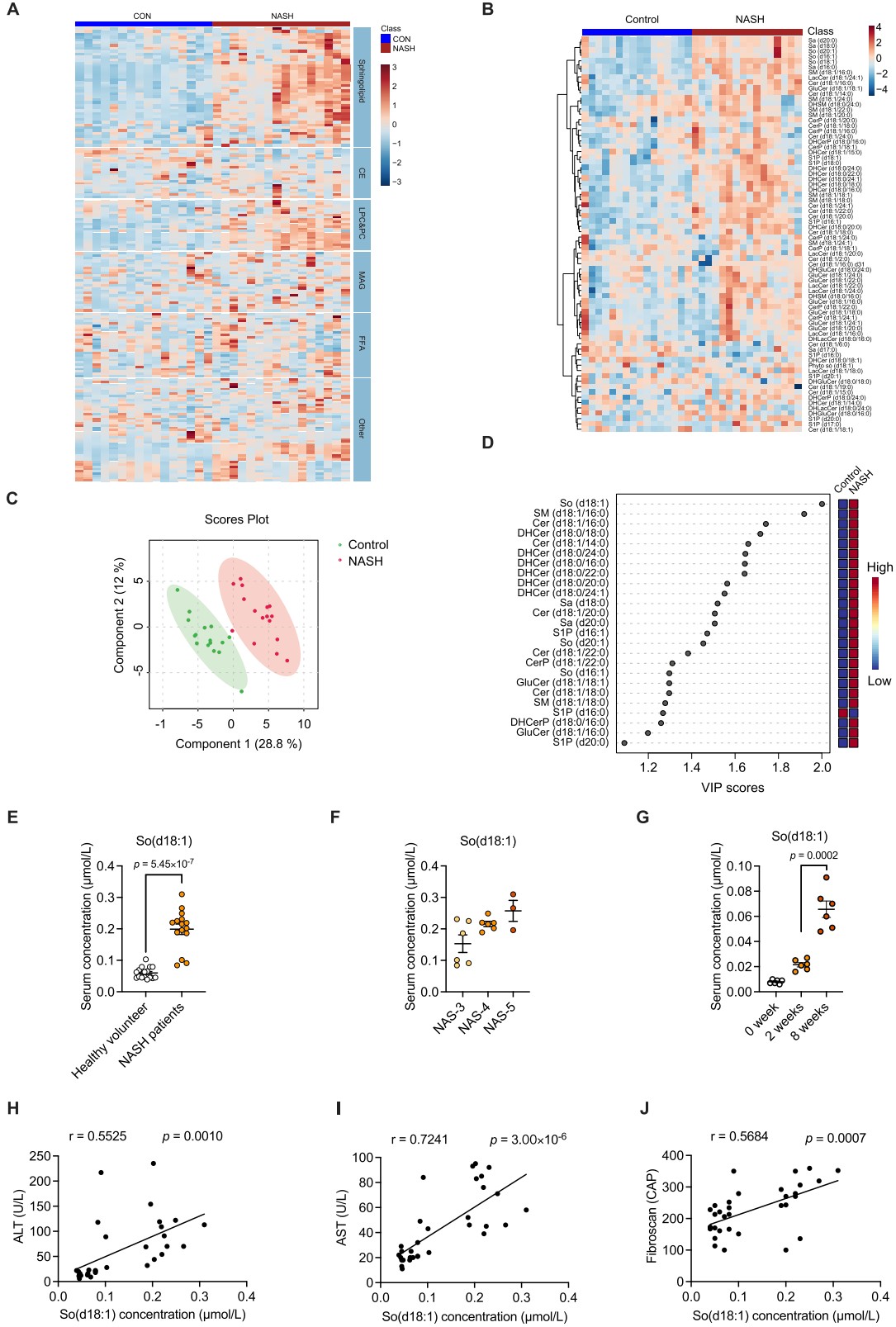

The GSEA enrichment analysis revealed transcriptional changes in the HIF-2α-regulated signalling pathway (Fig. 3D), while the HIF-1α signalling pathway was not changed (Fig. S3D). Flow cytometric analysis could also show that CDAA-HFD feeding suppresses HIF-2α expression in liver macrophages (Fig. S3F).

To validate the RNA-seq results, we treated mouse BMDMs with So(d18:1) and other sphingolipids as controls. The results showed that the transcription levels of the *Hif2α* gene were not changed, but its downstream gene *Vegf* decreased after So(d18:1) treatment (Fig. 3E). We also detected *Hif1α* and its downstream genes and their expression levels were unchanged (Fig. S3E). As for the protein levels of HIF-2α, the results showed that So(d18:1) treatment could significantly inhibit the protein expression of HIF-2α (Fig. 3F). In addition to the in vitro experiments, we also detected the expression of HIF-2α in liver

**Fig. 1 | Metabolomic analysis revealed changes in sphingosine in NASH patients.** Metabolic analysis of serum samples collected from NASH patients ($n$ = 16) and healthy control ($n$ = 16). **A** Clustering heatmap of the metabolic pathway, colours represent the intensity of response. **B** Targeted metabonomic detection of sphingolipids, colours represent the intensity of response. **C** PLS-DA analysis of sphingolipids in serum of patients. **D** VIP score plot of the difference sphingolipids between the two groups. **E, F** Serum So(d18:1) concentration in healthy volunteers ($n$ = 16 biologically independent samples) and NASH patients ($n$ = 15 biologically independent samples) (**E**) and different stages of NASH patients classified by NAS scores, NAS-3 ($n$ = 6 biologically independent samples), NAS-4 ($n$ = 6 biologically independent samples), NAS-5 ($n$ = 3 biologically independent samples) (**F**). **G** Serum concentration of So(d18:1) in different time points of NASH modelling mice ($n$ = 6 biologically independent animals). **H–J** Correlative analysis of So(d18:1) concentration in serum with ALT (Alanine aminotransferase) (**H**), AST (Aspartate aminotransferase) (**I**) and Fibroscan index (**J**). Correlations between variables were assessed by linear regression analysis. Linear correction index $R$ square and $P$-values were calculated. Data are the means ± s.e.m. Statistical analysis was performed using two-tailed Student's $t$-tests (**E**), One-way ANOVA with Dunnett's T3 post hoc test (**F**) and Kruskal–Wallis test with Dunn's test (**G**).

macrophages from CDAA-HFD diet fed mice treated with vehicle (5% CMC-Na) or So(d18:1) using flow cytometry, and the results showed that So(d18:1) significantly inhibited the expression of HIF-2α in liver macrophages (Fig. S3G).

Intrahepatic macrophages IL-1β and IL-18 secretion due to NLRP3 inflammasome activation is an important mechanism that promotes the progression of NASH[11]. Our previous study also found that macrophage HIF-2α could suppress NLRP3 inflammasome activation by inhibiting CPT1A[12]. Consistently, western blot analysis showed that So(d18:1) administration increased cleaved-caspase-1, indicating So(d18:1) could increase inflammasome assembly therefore increase Caspase-1 cleavage, while HIF-2α overexpression could quell the stimulation caused by So(d18:1) (Fig. 3G). IL-1β and IL-18 secretion levels also confirmed that only So(d18:1) promoted inflammasome activation (Fig. S3H, I), but not in HIF-2α overexpressing macrophages (Fig. 3H, I). Besides, So(d18:1) didn't damage the hepatocytes or influence the fibrosis-related genes in hepatic stellate cells directly (Fig. S3J, K).

These results indicate that So(d18:1) promotes the inflammatory factors secretion of macrophages through inhibition of HIF-2α, which may be the cellular mechanism by which So(d18:1) promotes hepatic inflammation in NASH.

## Macrophage-specific HIF-2α deletion aggravates inflammation and fibrosis in NASH

To investigate whether HIF-2α in macrophages can influence NASH disease progression, we fed $Hif2\alpha^{fl/fl}$ and $Hif2\alpha^{\Delta Lysm}$ mice a GAN diet (40% fat-powered, 20% fructose, and 2% cholesterol) for 24 weeks to compare the severity of inflammation and fibrosis in the liver. There was no significant difference in body weight between the two groups of mice (Fig. S4A). Liver weight and the ratio of liver weight to body weight were significantly increased in $Hif2\alpha^{\Delta Lysm}$ mice compared with $Hif2\alpha^{fl/fl}$ mice (Fig. 4A, B). Moreover, the levels of ALT and AST in the serum of $Hif2\alpha^{\Delta Lysm}$ mice were significantly higher than those in $Hif2\alpha^{fl/fl}$ mice, suggesting that knockdown of $Hif2a$ exacerbates the disease symptoms of NASH (Fig. 4C, D). Next, we examined the changes in lipids in the liver tissue and plasma of the two groups of mice. The results revealed that the concentrations of serum TG, CE, and NEFAs and hepatic TG and CE were not significantly different between $Hif2\alpha^{fl/fl}$ mice and $Hif2\alpha^{\Delta Lysm}$ mice (Fig. S4C–F). This result suggests that the knockdown of macrophage $Hif2a$ does not affect total lipid metabolism or consequently exacerbate lipid accumulation in the liver.

To further determine the changes in the levels of inflammation and fibrosis within the mouse liver to determine the progression of NASH, pathological sections were made from the livers of the two groups of mice to observe the extent of liver injury in the mice (Fig. 4E). The degree of hepatic steatosis was consistent between the two groups of mice (Fig. 4G), and there was no significant difference in the ballooning score (Fig. 4I). However, mice in the $Hif2\alpha^{\Delta Lysm}$ group had more foci of inflammation in the liver, with a large number of mononuclear macrophages diffusely distributed and a significantly higher inflammation score in the liver lobules than in the $Hif2\alpha^{fl/fl}$ group (Fig. 4H). The sections were also stained with Sirius red (Fig. 4E), and the fibrosis area was quantified to show that the $Hif2\alpha^{\Delta Lysm}$ group had a significantly greater fibrosis area than that of the $Hif2\alpha^{fl/fl}$ group (Fig. 4F). These

results demonstrate that macrophage $Hif2a$ knockdown can indeed significantly exacerbate NASH symptoms and promote inflammatory activation and fibrosis formation.

Consistently, the mRNA expression of inflammation genes and fibrosis genes was significantly upregulated in the livers of $Hif2\alpha^{\Delta Lysm}$ mice compared with that of $Hif2\alpha^{fl/fl}$ mice (Fig. 4K, L), while the lipid metabolism genes were not different between the two groups (Fig. S4G). Collectively, these data showed that genetic disruption of macrophage-specific HIF-2α accelerated hepatic inflammation and fibrosis but did not affect hepatic steatosis.

## Macrophage-specific HIF-2α overexpression alleviated inflammation and fibrosis in NASH and confronted the effect of So(d18:1) on NASH

To further verify the role of macrophage HIF-2α overexpression in NASH and investigate whether macrophage-specific HIF-2α overexpression could confront the effects of So(d18:1) on NASH in vivo, $Hif2\alpha^{+/+}$ and LysM$Hif2\alpha^{LSL/LSL}$ mice were fed a CDAA-HFD with vehicle or So(d18:1) injection for 8 weeks. There was no significant difference in body weight (Fig. S5A). The liver weight and the ratio of liver weight to body weight decreased in LysM$Hif2\alpha^{LSL/LSL}$ mice compared with $Hif2\alpha^{+/+}$ mice (Fig. 5A, B).

The levels of ALT and AST in the serum were significantly lower in LysM$Hif2\alpha^{LSL/LSL}$ mice than in $Hif2\alpha^{+/+}$ mice (Fig. 5C, D), suggesting that $Hif2a$ overexpression can protect the liver and reduce liver injury. Moreover, macrophage-specific HIF-2α overexpression lowered the ALT and AST lever in LysM$Hif2\alpha^{LSL/LSL}$ + So(d18:1) group compared with $Hif2\alpha^{+/+}$ + So(d18:1) group (Fig. 5C, D). We also measured TG, total CE and NEFA levels in the liver and plasma to investigate whether macrophage-specific $Hif2a$ overexpression could reduce fat accumulation in the liver, but there were no differences in any of these parameters (Fig. S5B–F).

The liver tissues of the four groups of mice were also paraffin sectioned and stained with H&E and Sirius red. The H&E staining results showed that there was no significant difference in the steatosis scores and hepatocyte ballooning scores (Fig. 5E, G, I). However, the LysM$Hif2\alpha^{LSL/LSL}$ group mice had fewer inflammatory foci in the liver, so their hepatic lobular inflammation scores were significantly lower than those of the $Hif2\alpha^{+/+}$ group mice (Fig. 5H) and the final calculated NAS of the LysM$Hif2\alpha^{LSL/LSL}$ group mice was significantly lower than that of the $Hif2\alpha^{+/+}$ group mice (Fig. 5J). We next examined Sirius red-stained sections, and it was evident that intrahepatic fibrosis production was reduced in the LysM$Hif2\alpha^{LSL/LSL}$ group of mice (Fig. 5E, F). So(d18:1) significantly increased the lobular inflammation, the calculated NAS and the fibrosis area in $Hif2\alpha^{+/+}$ mice, but not in LysM$Hif2\alpha^{LSL/LSL}$ mice (Fig. 5E–J). Importantly, the lobular inflammation scores, NAS and fibrosis area increased by So(d18:1) treatment were alleviated by the overexpression of HIF2a as comparing LysM$Hif2\alpha^{LSL/LSL}$ + So(d18:1) group with $Hif2\alpha^{+/+}$ + So(d18:1) group (Fig. 5E–J).

Consistently, the mRNA expression of inflammation-related genes and fibrosis-related genes was significantly downregulated in the livers of LysM$Hif2\alpha^{LSL/LSL}$ mice compared with $Hif2\alpha^{+/+}$ mice (Fig. 5K, L), while the lipid metabolism-related genes were not different (Fig. S5G). So(d18:1) increased the mRNA level of inflammation-related genes and

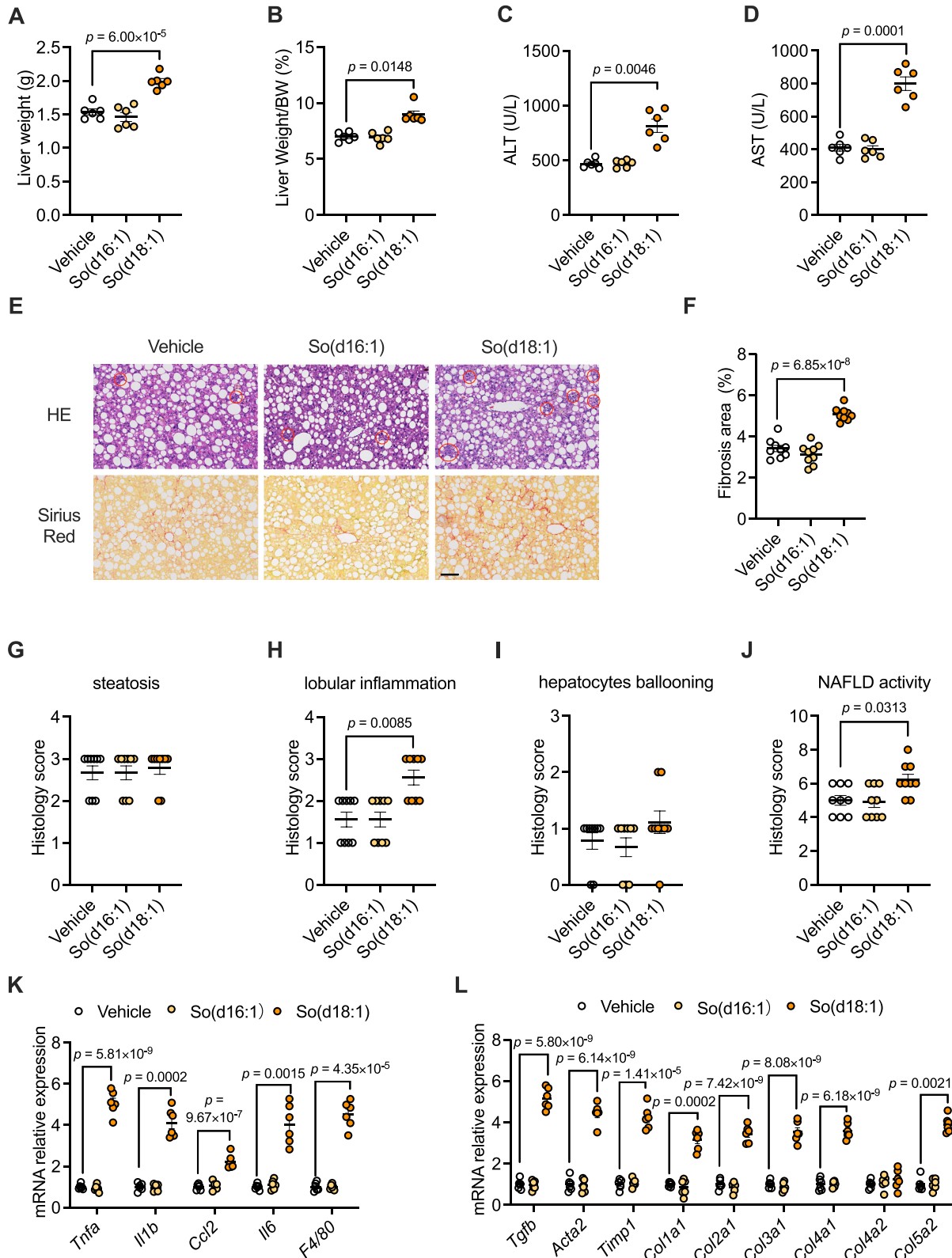

**Fig. 2 | Sphingosine 18:1 aggravates NASH.** CDAA-HFD-fed mice were treated with vehicle, sphingosine 16:1 or sphingosine 18:1 for 8 weeks (*n* = 6 mice/group). **A** Liver weights. **B** Ratios of liver mass to body mass. **C** Serum ALT. **D** Serum AST. **E** Representative H&E (up), and Sirius Red (down) staining of liver sections. The circles marked the inflammation foci. *n* = 3 mice per group, 3 images per mouse. Scale bar is 100 μm. **F** The percentage of fibrosis area. **G**–**J** Histology scores of hepatic steatosis (**G**), lobular inflammation (**H**), ballooning (**I**), and NAFLD activity (**J**). **K**, **L** Relative mRNA levels of genes related to hepatic inflammation (**K**) and fibrosis (**L**). *n* = 6 biologically independent samples. Data are the means ± s.e.m. **A**, **C**, **D**, **F**, **K**, **L** Statistical analysis was performed using One-way ANOVA; **B**, **G**–**J** *Col5a2* in L, statistical analysis was performed using Kruskal–Wallis test with Dunn's test.

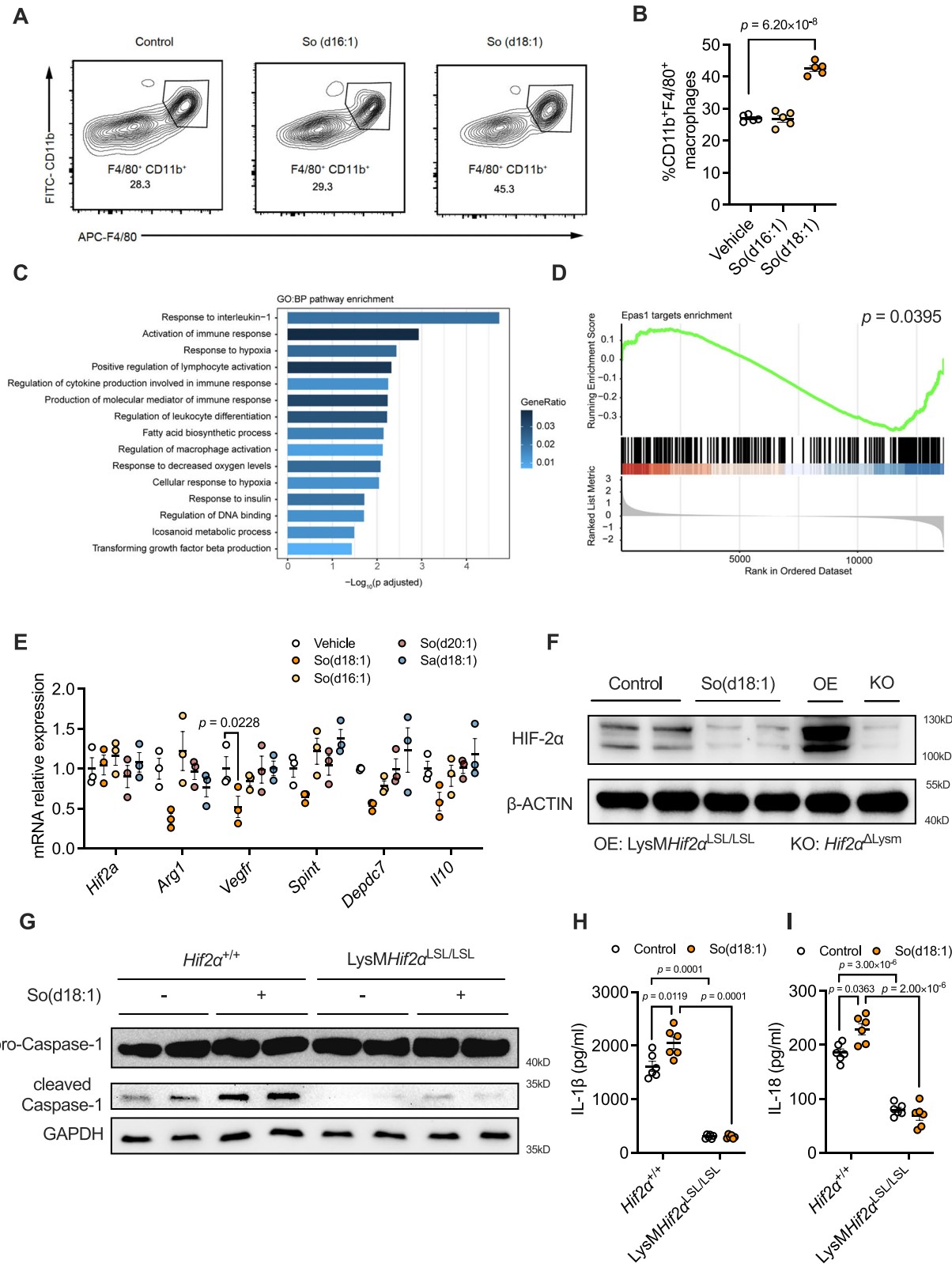

fibrosis in $Hif2\alpha^{+/+}$ mice, but not in LysM$Hif2\alpha^{LSL/LSL}$ mice (Fig. 5K, L). Moreover, the mRNA expression of inflammation-related genes and fibrosis-related genes were reduced by the overexpression of HIF2a as comparing LysM$Hif2\alpha^{LSL/LSL}$ + So(d18:1) group with $Hif2\alpha^{+/+}$ + So(d18:1) group (Fig. 5K, L). Collectively, these data showed that macrophage-specific HIF-2α overexpression ameliorated hepatic inflammation and fibrosis but did not affect hepatic steatosis. Moreover, macrophage-

specific HIF-2α overexpression confronted the effects of So(d18:1) on NASH in vivo.

**So(d18:1) reduces the transcriptional activity of HIF-2α**
In previous results, we have verified that So(d18:1) could promote NLRP3 inflammasome activation in macrophages and identified HIF-2α as a key transcription factor by which So(d18:1) alters the inflammatory

**Fig. 3 | So(d18:1) inhibits HIF-2α transcription function in liver macrophages.**
**A**, **B** Flow cytometric and statistical analysis of chow-diet fed mice treated with control (5% CMC-Na), So(d16:1) and So(d18:1) (*n* = 5 per group). **C** GO:BP pathway enrichment showing the transcriptional level changes of some immune-related pathways. (*n* = 4 biologically independent samples). **D** *Epas1* targets enrichment. **E** Relative mRNA levels of *Hif2α* and its downstream target genes in macrophages treated with vehicle, So(d18:1), So(d16:1), So(d20:1) or Sa(d18:1). (*n* = 3). **F** Representative immunoblot analysis of HIF-2α of BMDMs isolated from wild-type mice stimulated with the vehicle and So(d18:1) and BMDMs isolated from LysM*Hif2α*[LSL/LSL] and *Hif2α*[△LysM] BMDMs stimulated with the vehicle. This experiment was repeated 3 times independently with similar results. **G** Representative

immunoblot analysis of pro-caspase-1 and caspase-1 from *Hif2α*[+/+] and LysM*Hif2α*[LSL/LSL] BMDMs that were treated with So(d18:1) or not under NLRP3 inflammasome stimulation. This experiment was repeated 3 times independently with similar results. **H**, **I** Protein level of IL-1β (**H**), IL-18 (**I**) from *Hif2α*[+/+] and LysM*Hif2α*[LSL/LSL] BMDMs treated with So(d18:1) or not under NLRP3 inflammasome stimulation. (*n* = 6 biologically independent samples). Data are the means ± s.e.m. **B**, **E**, **H**, **I** Statistical analysis was performed using One-Way ANOVA test; *Spint* in (**E**), statistical analysis was performed using the Kruskal–Wallis test. **C** One-sided hypergeometric distribution test, using Benjamin Hochberg method for *p*-value correction. **D** One-sided permutation test without *p*-value correction.

state of macrophages. Therefore, how does the increased So(d18:1) in NASH patients affect HIF-2α protein function in macrophages? We conducted a more in-depth mechanistic study to address this question. To determine whether So(d18:1) could inhibit HIF-2α transcriptional activity, we constructed a HIF response element (HRE)-based luciferase reporter assay. Fluorescein detection showed that So(d18:1) could significantly inhibit the transcriptional activity of HIF-2α (Fig. 6A). But So(d18:1) couldn't inhibit HIF-1α transcription (Fig. 6B). The transcriptional action of HIF-2α requires binding to the ARNT subunit. Thus, we utilized a mammalian two-hybrid system that could further verify that So(d18:1) repressed the transcriptional function of HIF-2α by inhibiting the binding of ARNT (Fig. 6C). In addition, co-immunoprecipitation revealed that So(d18:1) disrupts the binding of HIF-2α to ARNT (Fig. 6D), but doesn't disrupt the direct binding of HIF-1α to ARNT (Fig. 6E).

HIF-2α has a hydrophobic pocket PAS-B domain to bind with ARNT[13]. Structural prediction by docking revealed some potential for So(d18:1) to fill into this hydrophobic pocket (Fig. 6F). We then constructed a HIF-2α plasmid with two proven missense mutations (S304M and G323E) in the pocket which disabled other molecules to bind with HIF-2α[14]. Luciferase reporter system was performed, and the results showed that So(d18:1) could normally inhibit the binding of wild-type HIF-2α to ARNT but not that of mutant HIF-2α to ARNT (Fig. 6G). From these results, we learned that So(d18:1) may fill into the hydrophobic pocket of HIF-2α and thereby inhibit the binding of HIF-2α to ARNT, which impedes HIF-2α entry into the nucleus for transcriptional regulation. HIF-2α that remains in the cytoplasm is very easily hydrolysed and therefore protein levels are reduced. This finding also explained why So(d18:1) can only change the protein expression level of HIF-2α but not the mRNA expression level.

HIF-2α regulates metabolism reprogramming and NLRP3 inflammasome activation by binding to the rHRE region on the *Cpt1a* promoter[12]. We therefore transfected a luciferase reporter gene plasmid containing a *Cpt1a* rHRE region with a HIF-2α plasmid or empty plasmid into cells, treated with control solvent or So(d18:1) and observed the fluorescence activity ratio. The results showed that in the vehicle group, the fluorescence values of HIF-2α[TM]-transfected cells were lower than those with empty plasmid, indicating that overexpression of HIF-2α inhibited the transcription of *Cpt1a* rHRE-linked luciferase. In contrast, the overexpression of HIF-2α in the So(d18:1) group did not affect the transcription of *Cpt1a* rHRE-linked luciferase, as it was unable to bind to ARNT and localize to the rHRE region in the nucleus, so the fluorescence values of HIF-2α[TM] plasmid-transfected cells were similar to the fluorescence values of the empty plasmid group (Fig. 6H).

The above results suggest that So(d18:1) could inhibit the binding of HIF-2α with ARNT to inhibit the transcriptional activity of HIF-2α, thus influencing the expression of downstream genes regulated by HIF-2α. These results also suggest the possibility that lipids bind directly to transcription factors and regulate their functions, showing us an interesting mechanism by which lipids influence cellular metabolism.

## Stabilization of HIF-2α expression in macrophages significantly alleviated inflammation and fibrosis in NASH

We further investigated the therapeutic effect of the specific HIF-2α agonist FG-4592 in treating NASH. SPF mice were given a CDAA-HFD diet for 8 weeks and were administered vehicle or FG-4592 (25 mg/kg) by intraperitoneal injection. At the end of the treatment, there was no significant difference in body weight between the two groups of mice (Fig. S6A), but there was a significant reduction in liver weight (Fig. 7A), as well as a significant reduction in the calculated liver weight/body weight ratio (Fig. 7B). Measurement of the blood levels of ALT and AST showed significant decreases in both transaminase levels suggestive of liver injury (Fig. 7C, D). To assess whether FG-4592 could improve intrahepatic fat accumulation, we also measured intrahepatic TG (Fig. S6B) and blood TG levels (Fig. S6C), neither of which showed a significant change. Total intrahepatic CE (Fig. S6D) and total plasma CE levels (Fig. S6E) were also tested, and there was no significant improvement in either of these results. With respect to NEFAs in the blood, there was also no improvement after FG-4592 injection (Fig. S6F). The above results suggest that although FG-4592 may improve liver injury, it does not improve lipid accumulation in the liver.

To further observe liver injury in mice, we made paraffin sections of liver tissue from both groups and stained them with H&E and Sirius red. In the H&E-stained sections, we observed that the degree of steatosis in the livers of the two groups of mice was the same, and therefore, there was no difference in the steatosis score and hepatocyte ballooning (Fig. 7G, I). However, there were significantly fewer foci of inflammation than in the vehicle group, and therefore, the score of the lobular inflammation was lower than that of the vehicle group (Fig. 7H). The final calculation of the NASs also showed that FG-4592 injection reduced the symptoms of NASH in mice (Fig. 7J). In Sirius red-stained sections, fibrosis was significantly reduced in the FG-4592-injected group, and this result could be better visualized by counting the fibrosis area proportion (Fig. 7E, F).

After FG-4592 injection, inflammation-related genes were significantly downregulated in the mouse liver (Fig. 7K). Additionally, genes related to fibrosis were significantly reduced (Fig. 7L). However, genes related to fatty acid uptake and de novo synthesis were slightly changed, with only the expression level of *Fasn* being reduced (Fig. S6G).

In conclusion, FG-4592 injection can reduce liver fibrosis and improve NASH symptoms by reducing intrahepatic inflammation.

## Discussion

Chronic liver injury caused by NASH can significantly increase the risk of end-stage liver diseases. However, there is currently no effective drug to treat NASH in the clinical practice. Here, we found that the abundance of So(d18:1) in patients with NASH was significantly increased through metabolomics analysis. So(d18:1) significantly aggravated hepatic lobular inflammation and fibrosis in the livers of NASH model mice. Mechanistically, So(d18:1) inhibits macrophage HIF-2α binding with ARNT, thus increasing the secretion of inflammatory factors. This mechanism reveals that macrophage HIF-2α may be a

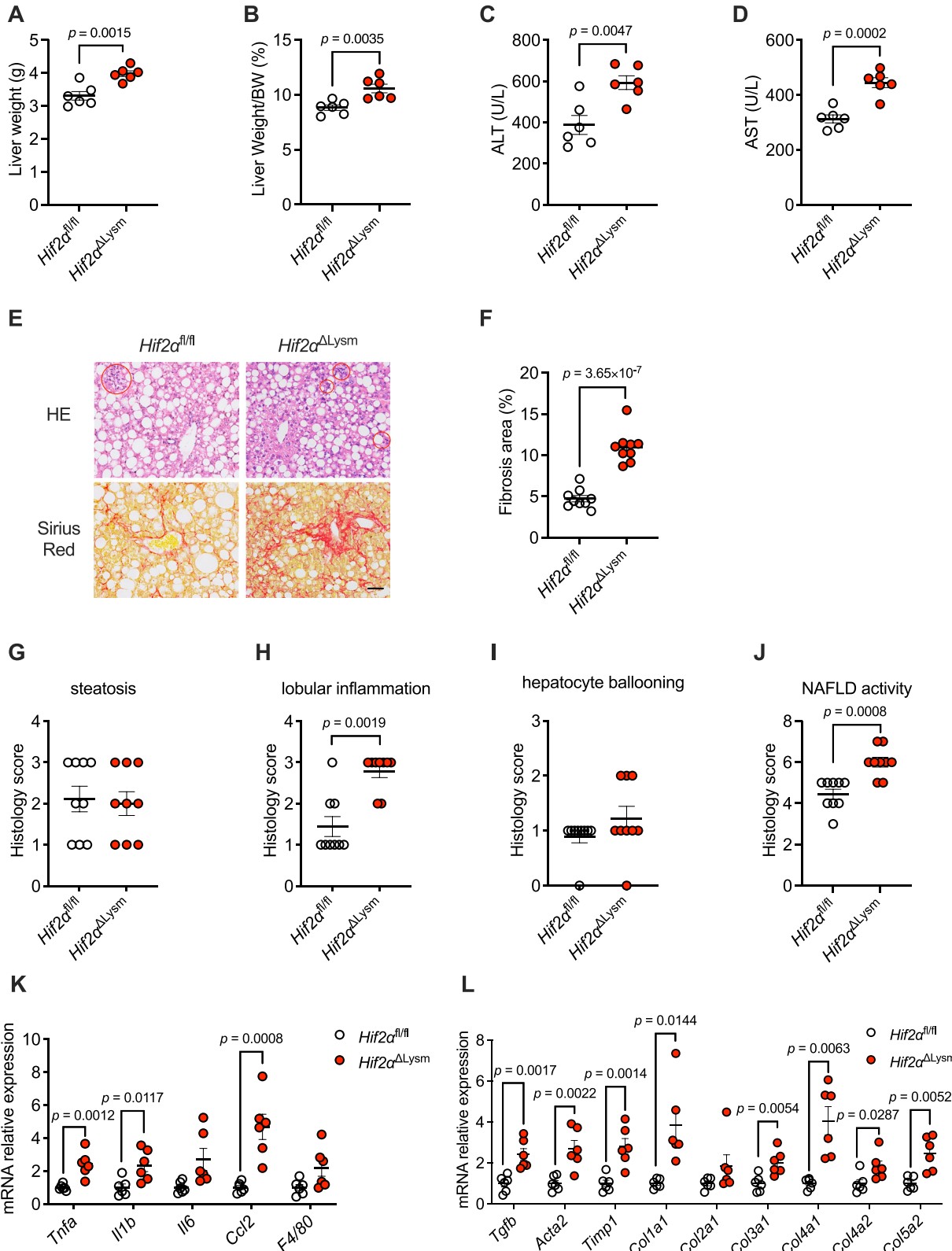

**Fig. 4 | HIF-2α KO in macrophages accelerated inflammation and fibrosis in NASH mice.** Eight-week-old male *Hif2α*^fl/fl^ and *Hif2α*^ΔLysm^ mice were administered a GAN diet for 24 weeks (SPF, *n* = 6 mice/group). **A** Liver weights. **B** Ratios of liver mass to body mass. **C** Serum ALT. **D** Serum AST. **E** Representative H&E (up), and Sirius Red (down) staining of liver sections. The circles marked the inflammation foci. *n* = 3 mice per group, 3 images per mouse. Scale bar is 200 μm. **F** The percentage of fibrosis area. **G–J** Histology scores of hepatic steatosis (**G**), lobular inflammation (**H**), ballooning (**I**), and NAFLD activity (**J**). **K, L** Relative mRNA levels of genes related to hepatic inflammation (**K**) and fibrosis (**L**). *n* = 6 biologically independent samples. Data are the means ± s.e.m. **A–D, F, K, L** Statistical analysis was performed using two-tailed Student's *t*-tests; **G–J** *Col2a1* in (**L**), statistical analysis was performed using two-tailed Mann–Whitney *U*-tests.

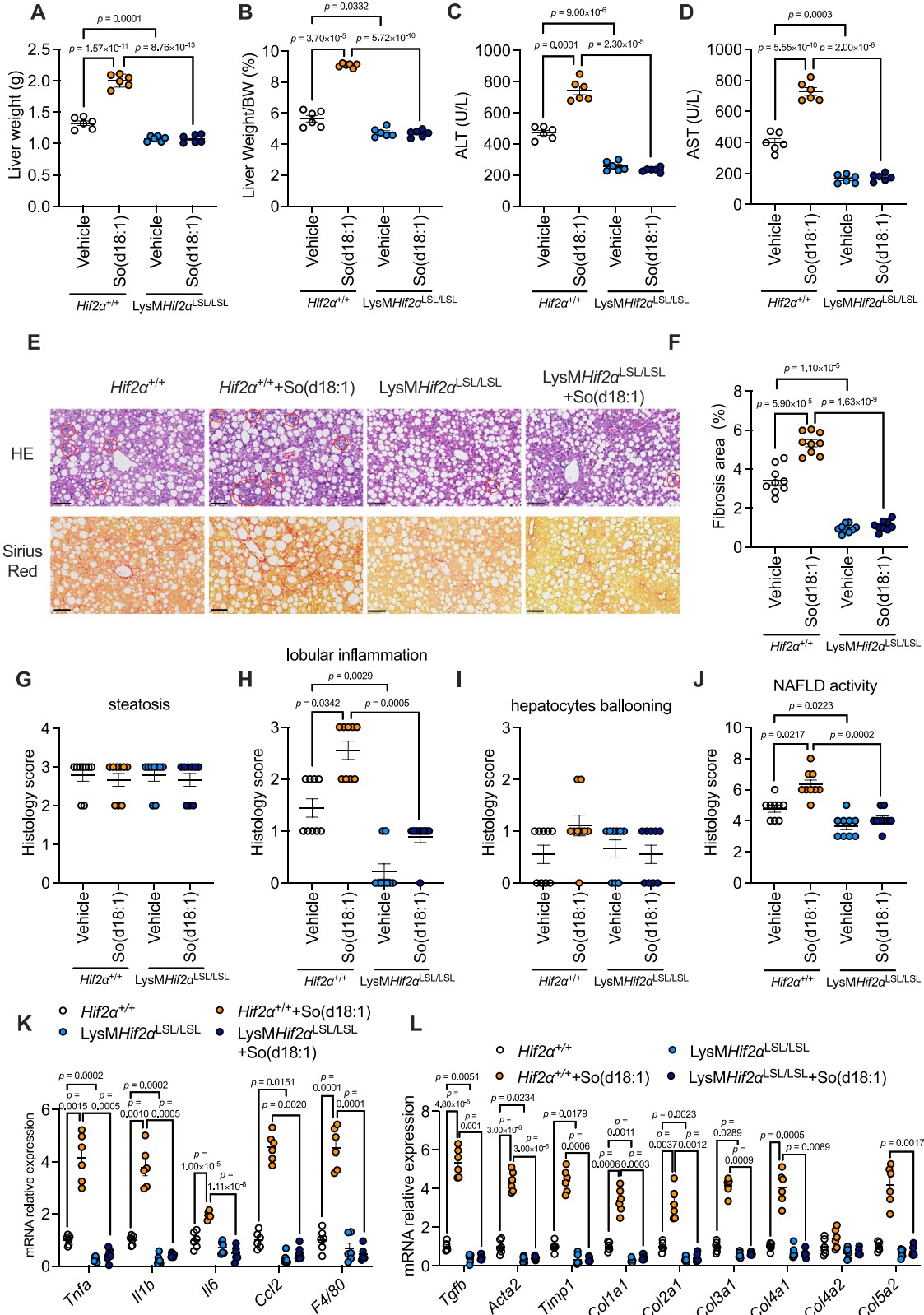

**Fig. 5 | HIF-2α overexpression in macrophages ameliorated inflammation and fibrosis in NASH mice.** Eight-week-old male *Hif2α*[+/+] and LysM*Hif2α*[LSL/LSL] mice treated with or without So(d18:1) by daily intraperitoneal injection under CDAA-HFD for 8 weeks (SPF, *n* = 6 mice/group). **A** Liver weights. **B** Ratios of liver mass to body mass. **C** Serum ALT. **D** Serum AST. **E** Representative H&E (up), and Sirius Red (down) staining of liver sections. The circles marked the inflammation foci. *n* = 3 mice per group, 3 images per mouse. Scale bar is 100 μm. **F** The percentage of

fibrosis area. **G**–**J** Histology scores of hepatic steatosis (**G**), lobular inflammation (**H**), ballooning (**I**), and NAFLD activity (**J**). **K** Relative mRNA levels of genes related to hepatic inflammation. **L** Relative mRNA levels of genes related to hepatic fibrosis. *n* = 6 biologically independent samples. Data are the means ± s.e.m. **A**–**D**, **F**, **K**, **L** Statistical analysis was performed using one-way ANOVA; **G**–**J** *Ccl2* in (**K**), *Timp1*, *Col3a1*, *Col4a2* and *Col5a2* in (**L**), statistical analysis was performed using the Kruskal–Wallis.

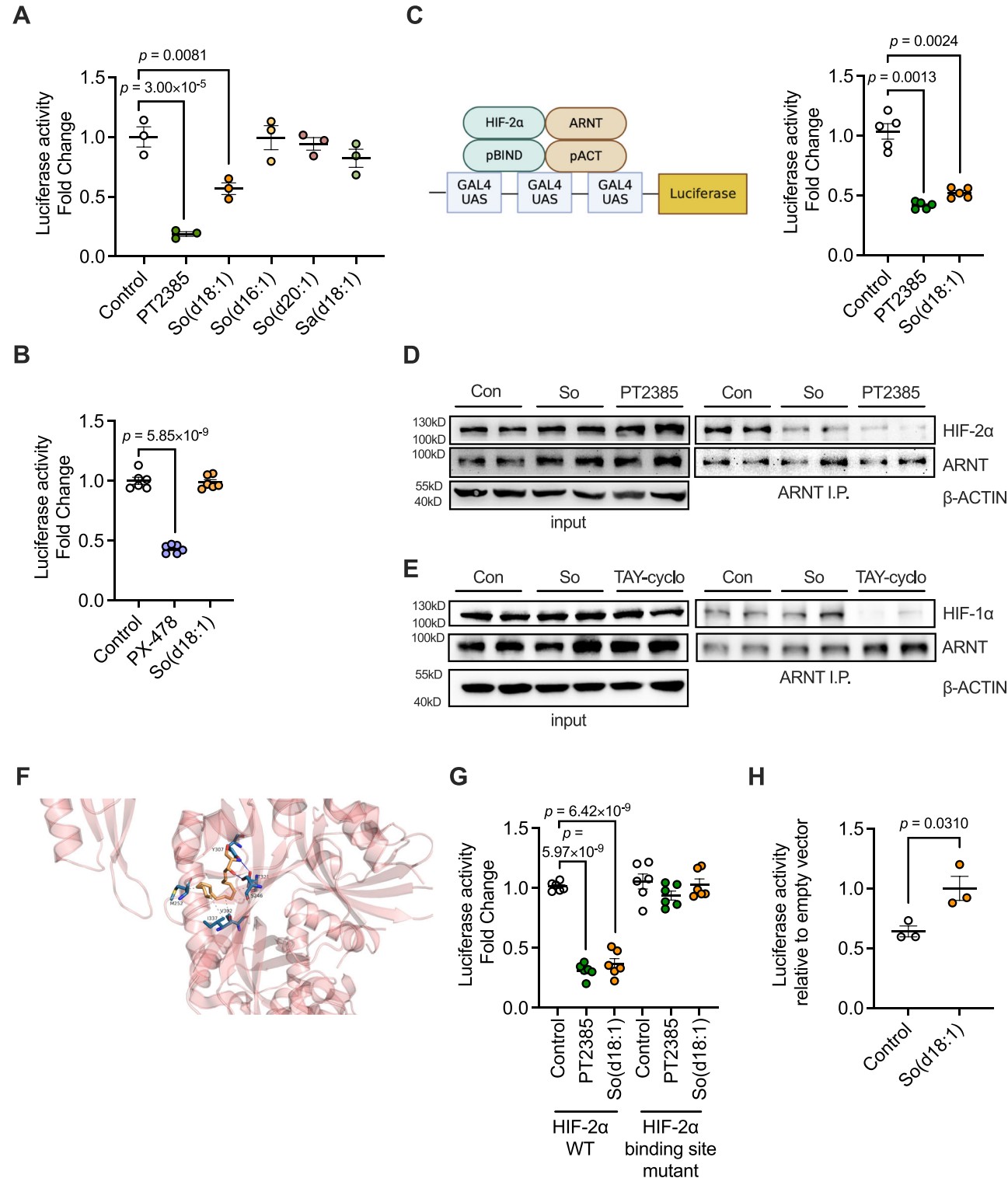

potential target for the treatment of NASH. Based on this finding, we tried to use the HIF-2α stabilizer FG-4592 to improve NASH, and the results showed that FG-4592 alleviated inflammation and fibrosis in NASH.

Liver steatosis is an early event of NASH. A large amount of lipid accumulation in hepatocytes leads to excessive oxidative stress in hepatocytes, which further induces hepatocyte death, thereby activating inflammation and fibrosis in hepatic lobules[4]. In NASH patients, we have seen several significant changes in sphingolipids, such as Cer (d18:1/16:0), Cer (d18:1/14:0), Cer (d18:1/20:0), Cer (d18:1/22:0), Cer

(d18:1/18:0) and Cer (d18:1/18:0). Their abundances also increased in the serum of NASH patients. Our previous work found that the excessive accumulation of nicotine in the intestine can promote the secretion of intestinal ceramide, thus promoting the progression of NAFLD to NASH[6]. Other studies also found that increasing liver Cer (d18:1/18:0), and S1P could limit the occurrence and development of chronic inflammation in NAFLD[15,16]. These studies suggest that sphingolipid metabolism may play an important role in the pathogenesis of NAFLD. However, none of these sphingolipids' changes could perfectly fit the trend of NASH disease exacerbation and indicate the severity of

**Fig. 6 | So(d18:1) suppress the binding of HIF-2α and ARNT. A** HRE-based luciferase assay in HEK293T transfected with HIF-2α, followed by treatment of PT2385, So(d18:1), So(d16:1), So(d20:1), Sa(d18:1) (*n* = 3 biologically independent samples). **B** HRE-based luciferase assay in HEK293T transfected with HIF-1α, followed by treatment of PX-478, So(d18:1) (*n* = 6 biologically independent samples). **C** Schematic and experimental representation of PT2385 or So(d18:1) mediated HIF-2α-ARNT interaction by pG5GAL4 luciferase assay followed by pBIND-HIF-2α and pACT-ARNT transfection of HEK293T cells (*n* = 5). **D** Co-immunoprecipitation for ARNT and HIF-2α in HEK293T cells transfected with ARNT and the oxygen-stable HIF-2α triple mutant (HIF-2α™) plasmid treated with control solvent, So(d18:1) or PT2385, PT2385 and So(d18:1) could inhibit HIF-2α to bind to ARNT. This experiment was repeated 3 times independently with similar results. **E** Co-immunoprecipitation for ARNT and HIF-1α in HEK293T cells transfected with ARNT and the oxygen-stable HIF-2α triple mutant (HIF-2α™) plasmid treated with So(d18:1) (20 µM) or TAY-cyclo (20 µM). This experiment was repeated 3 times independently with similar results. **F** Molecule docking prediction of So(d18:1) binding sites in HIF-2α PAS-B domain. **G** schematic diagram of site missense mutation experiment. PT2385 and So(d18:1) could inhibit normal HIF-2α transcription ability but not HIF-2α with missense mutations. (*n* = 6 biologically independent samples). **H** *Cpt1a* promoter rHRE constructs plasmid were co-transfected with HIF-2α™ followed by control solvent or So(d18:1) treatment. So(d18:1) could inhibit HIF-2α binding ability to rHRE. (*n* = 3 biologically independent samples). Data are the means ± s.e.m. **A–C**, **G** (WT), statistical analysis was performed using One-way ANOVA. G(mutant), statistical analysis was performed using Kruskal–Wallis test with Dunn's test. **H** Statistical analysis was performed using two-tailed Student's *t*-tests.

NASH. But So(d18:1) showed a close relationship to the disease progression of NASH by completely consistent with the trends of the changes in ALT and AST levels representing liver injury. Thus, we think So(d18:1) may be a better indicator of the progression of NASH.

Although sphingosine hasn't been deeply discussed in many studies, there still are some studies that have found the sphingosine's shape in metabolomics[7,8], and they have even found that So(d18:1) in stool can be used as a biomarker to predict cirrhosis[9]. However, So(d18:1) is usually regarded just as an intermediate product of metabolism between ceramide and S1P, and in-depth mechanistic and functional research is lacking. In this study, we found that So(d18:1) can exist stably in cells at a certain concentration and will not be rapidly converted into ceramide or S1P. Our results showed that So(d18:1) can not only promote the secretion of inflammatory factors in BMDMs but can also aggravate liver inflammation and fibrosis and promote the progression of NASH in animals. Besides, we found that So(d18:1) didn't directly promote hepatocyte death, nor did it directly induce fibrosis in LX-2 cells. So(d18:1) may still have the possibility of directly acting on other liver cell types. However, myeloid-specific HIF-2α overexpression confronted the effects of So(d18:1) on NASH largely (Fig. 5), which could demonstrate the central role of macrophages in this process.

Regarding the origin of the increased circulating So(d18:1) in NASH patients, we examined the amount of So(d18:1) in the whole liver tissue of NASH-modelled mice (Fig. S1C). It showed a downward trend within the first two weeks, but no further change in the coming 6 weeks. The metabolism of ceramide is also known to occur in the gut and adipose tissue. There are also results showing that increased levels of So(d18:1) in the feces of NASH-cirrhotic patients, which may serve as one of the biomarkers for predicting NASH-cirrhosis[9]. This also suggests that the role of microbiota in sphingolipid metabolism should not be underestimated. In summary, the increased So(d18:1) may come from damaged hepatocytes and other organs, especially the gut. Next, we will subsequently measure the levels of So(d18:1) in the gut and adipose tissue of NASH-modelled mice at different time points to further investigate the source of the increased circulating So(d18:1).

HIF is a heterodimer made up of an oxygen-sensitive α subunit and a constitutively expressed β subunit (ARNT). Under normoxic conditions, HIF-α is rapidly hydroxylated and degraded by prolyl hydroxylase (PHD). In contrast, under hypoxia, the activity of prolyl hydroxylase was inhibited, and the HIF protein was stable. HIF-2α accumulates and translocates to the nucleus and combines with ARNT to form an active transcription factor complex[17]. In NASH, HIF-1α in macrophages induced by palmitic acid damages autophagic flux and increases IL-1β production, aggravating liver injury induced by an MCD diet[18]. However, the role of HIF-2α in macrophages in the progression of NASH is still unclear. In this article, we have validated the role of HIF-2α in NASH progression using mice with macrophage-specific knockdown or overexpression of HIF-2α. Identified that HIF-2α is a potential target for intervention in NASH.

Rosalistat (FG-4592) is a mature small-molecule drug that is mainly used to treat chronic kidney disease and anaemia, but its role in metabolic diseases has not yet entered clinical trials. In our previous studies, FG-4592 injection was used to improve insulin resistance[12]. In this study, FG-4592 injection significantly reduced the levels of ALT and AST in the livers of mice, suggesting that the degree of liver injury was reduced. In addition, the expression of genes related to inflammation and fibrosis also decreased. The above results showed that FG-4592 injection can reduce the incidence of NASH. However, many articles have also clarified that the overexpression of HIF-2α in liver cells plays a worsening role in insulin resistance and fatty liver[19,20]. Continuous activation of hepatocyte HIF-2α can damage the transcription of fatty acid β-oxidation-related genes, leading to fat accumulation in the liver[20,21]. Hepatocyte HIF-2α stimulated the production of histidine-rich glycoprotein (HRGP) to activate macrophages to polarize to the M1 type, thus causing liver damage. In our study, the administration of FG-4592 can block the response-ability of proinflammatory macrophages, thus playing a protective role. After FG-4592 reaches the liver, it may indeed lead to the accumulation of lipids in the liver, but it also ensures that hepatocytes damaged by lipotoxicity will not cause further macrophage inflammation. Therefore, from the overall animal experimental data, the administration of FG-4592 still protects the liver from damage in NASH disease. In addition, FG-4592 can also act on other targets, such as adipose tissue HIF-2α, and promote the production of erythropoietin[22,23], which will delay or even improve the disease in many chronic metabolic diseases. Of course, we are also actively seeking ways to improve FG-4592 drug delivery methods, such as using liposome encapsulation to minimize the side effects induced by FG-4592 activation of hepatocyte HIF-2α.

In summary, our study found that the active sphingolipid So(d18:1) has a good indicating ability in patients with NASH and that it can bind to HIF-2α to aggravate liver inflammation and fibrosis in NASH model mice. Macrophage-specific knockout or overexpression of HIF-2α showed that macrophage HIF-2α can reduce liver injury and can reduce intrahepatic inflammation and fibrosis. These results not only provide us with a possibility that So(d18:1), a long-chain lipid, binding transcription factor to regulate cellular immune metabolism, but also suggest that the proinflammatory function of So(d18:1) in NASH cannot be ignored. Finally, we used FG-4592 to improve inflammation and fibrosis in NASH. This study provides potential therapeutic targets and strategies for NASH.

## Methods

The use of patient samples in this research complies with the Ethics Committee of Peking University People's Hospital (Ethics Review Approval No.: 2021PHB124-001). All animal experiments complied with the rules for the use of experimental animals, treatment and euthanasia approved by Peking University Health Science Center (permit: LA2020481).

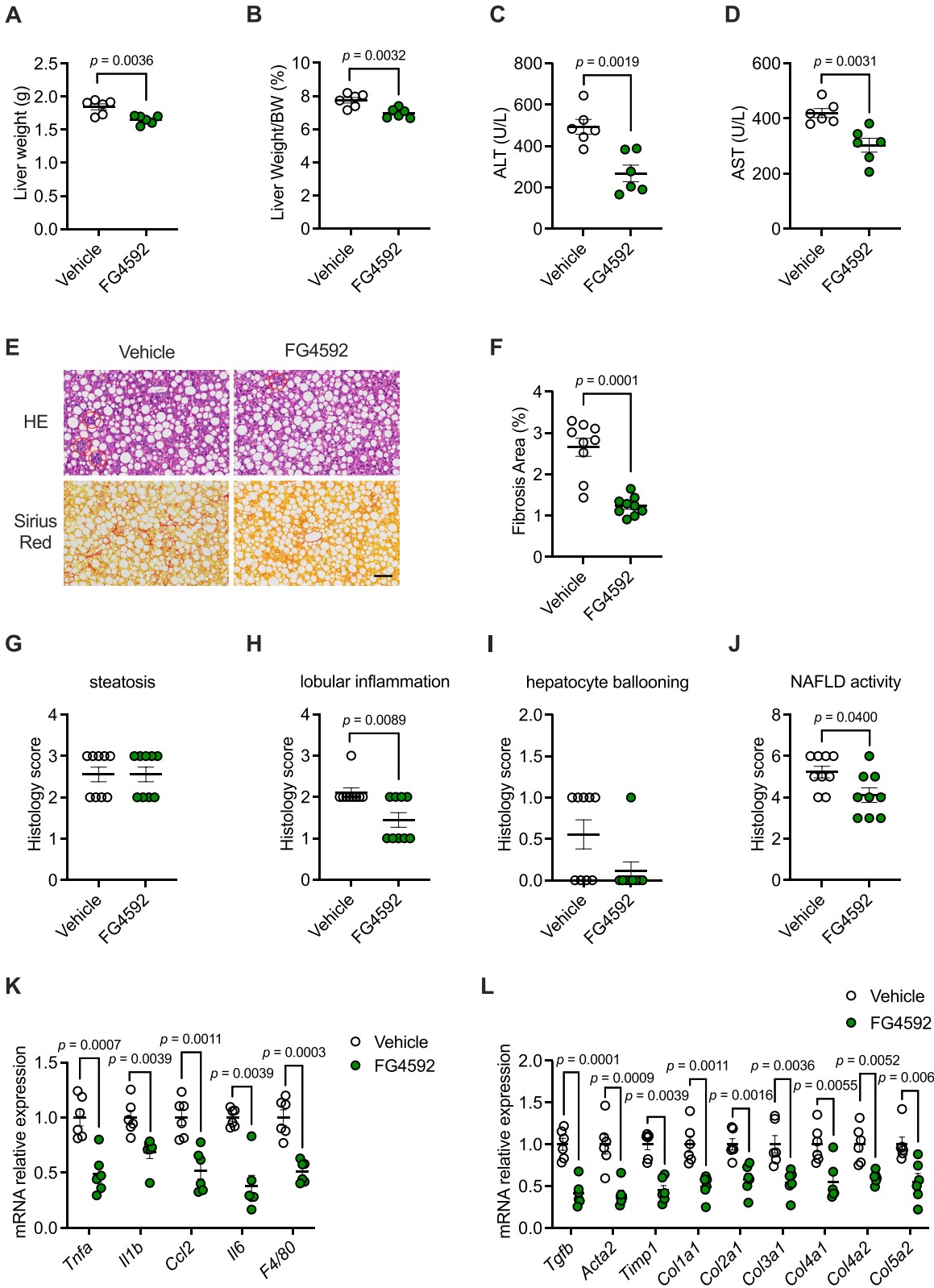

**Fig. 7 | FG-4592 significantly mitigates CDAA-HFD diet-induced NASH.** CDAA-HFD-fed mice were treated with vehicle or FG-4592 for 8 weeks (*n* = 6 mice/group). **A** Liver weights. **B** Ratios of liver mass to body mass. **C** Serum ALT. **D** Serum AST. **E** Representative H&E (up), and Sirius Red (down) staining of liver sections. The circles marked the inflammation foci. *n* = 3 mice per group, 3 images per mouse. Scale bar is 100 μm. **F** The percentage of fibrosis area. **G–J** Histology scores of hepatic steatosis (**G**), lobular inflammation (**H**), ballooning (**I**), and NAFLD activity (**J**). **K**, **L** Relative mRNA levels of genes related to hepatic inflammation (**K**) and fibrosis (**L**). *n* = 6 biologically independent samples. Data are the means ± s.e.m. **A–D**, **F**, **K**, **L** Statistical analysis was performed using two-tailed Student's *t*-tests; **G–J** *Il1b*, *Il6* in (**K**), *Timp1* and *Col5a2* in (**L**), statistical analysis was performed using two-tailed Mann–Whitney *U*-tests.

## Human participants

The clinical patient cohorts of this study were collected from Peking University People's Hospital. With the approval of the Ethics Committee of Peking University People's Hospital (Ethics Review Approval No.: 2021PHB124-001), all volunteers who participated in the study signed a written informed consent form.

The inclusion criteria were as follows: NASH disease diagnosis was in accordance with the *Guidelines of Prevention and Treatment of Non-Alcoholic Fatty Liver Disease: a 2018 Update* prepared by the National Workshop on Fatty Liver and Alcoholic Liver Disease, Chinese Society of Hepatology, Chinese Medical Association; Fatty Liver Experts Committee, Chinese Medical Doctor Association. The diagnosis requires the patient to have histological evidence of diffuse hepatocyte steatosis, intrahepatic inflammation and fibrosis, and persistent serum ALT and GGT increases. Patients with alcoholic liver disease, type 3 hepatitis C virus infection, autoimmune hepatitis, hepatolenticular degeneration and drug-induced liver disease were excluded. A FibroScan liver elasticity test was performed to support the diagnosis. All patients were newly diagnosed with NASH and did not receive relevant treatment. Healthy volunteers were also recruited from Peking University People's Hospital. They were required to have normal serum ALT and GGT levels. FibroScan indicated that their liver elasticity was normal. Their age, sex and BMI were matched to those of NASH patients. This study includes males and females, and the gender of participants was determined based on self-report.

## Animals

C57BL/6J wild-type male mice were purchased from the Department of Laboratory Animal Science, Peking University Health Science Center. *Hif2α*[fl/fl], *Hif2α*[LSL/LSL] and Lyz2-cre mice which carrying Cre recombinase under the control of the Lyz2 promoter were purchased from Jackson Lab. Hif2α[fl/fl] or Hif2α[LSL/LSL] mice were bred with Lyz2-cre mice to generate *Hif2α*[ΔLysm] mice and LysM*Hif2α*[LSL/LSL] mice.

Mice were randomly divided into different groups and raised in cages under standard SPF laboratory conditions with free access to water and feed. The temperature was maintained at 21-24 °C, and the humidity was maintained at 40–70%. The light was on from 08:00 to 20:00. The animal use licence number was SYXK (Beijing) 2011-0039. All animal experiments complied with the rules for the use of experimental animals, treatment and euthanasia approved by Peking University Health Science Center (permit: LA2020481).

A normal chow-diet (NCD, 1016706476803973120) was purchased from Beijing Keaoxieli Feed Co., Ltd., in which fat supplies 20% of calories for energy. The GAN diet (D09100310) was purchased from Research Diets, USA, in which fat provides 40% of calories for energy (including palm oil), fructose provides 20% of calories for energy, and 2% cholesterol is added. Mice were fed the GAN diet for 24 weeks to create the NASH model. The CDAA-HFD (A06071302) was purchased from Research Diets, USA, in which fat supplies 60% of calories for energy, and the diet contains 0.1% methionine and does not contain any added choline. Mice were fed the CDAA-HFD for 8 weeks to create the NASH model.

For the So(d18:1) intraperitoneal injection experiment, the So(d16:1) and So(d18:1) were suspended in 0.5% Carboxymethylcellulose sodium (CMC-Na) solution. 6-week-old male mice were randomly fed the CDAA-HFD for 8 weeks with vehicle (5% CMC-Na), So(d16:1) (10 mg kg$^{-1}$ body weight) or So(d18:1) (10 mg/kg body weight) injected intraperitoneally every day. For the FG-4592 (MCE, HY-13426) intraperitoneal injection experiment, 6-week-old male mice were randomly fed the CDAA-HFD for 8 weeks with vehicle or FG-4592 (25 mg/kg body weight) injected intraperitoneally every day.

## Cell lines

The HEK293T cell line (catalogue number: GNHu44) and LX-2 cell line (catalogue number: SCSP-527) used in this study were purchased from the National Collection of Authenticated Cell Cultures.

## Primary mouse bone marrow-derived macrophage culture

Bone marrow-derived macrophages were isolated from the bone marrow of C57BL/6J wild-type mice, macrophage-specific knockout HIF-2α mice (*Hif2α*[ΔLysm]) and macrophage-specific overexpressing HIF-2α mice (LysM*Hif2α*[LSL/LSL]).

BMDMs were prepared as previously described[24]. The bone marrow collected from the femur and tibia of mice was inoculated on sterile petri dishes and cultured in RPMI 1640 containing 10% FBS, 100 units/ml penicillin, 100 mg/ml streptomycin and 10 ng/ml macrophage colony-stimulating factor (M-CSF) for 5–6 days. When activating the NLRP3 inflammasome, BMDMs were incubated with LPS (500 ng/ml, 4 h) and then were treated with nigericin (6.7 μM, 1 h).

## Separation of liver nonparenchymal cells

As mentioned earlier[25], primary hepatic macrophages were isolated from male mice by injecting type IV collagenase into the liver. Mice were anaesthetized with isoflurane and perfused through the portal vein. Krebs buffer was used to remove blood from the liver. Then, Krebs buffer supplemented with type IV collagenase was used for digestion. After digestion, the liver was collected and rinsed with RPMI 1640. The digested liver cell suspension was passed through a 70-μm cell filter (BD). The samples were centrifuged at 50 × g for 3 min, and the supernatant was retained. The cells were centrifuged at 1200 rpm for 10 min again to precipitate the nonparenchymal cells from the supernatant.

## Flow cytometry

Isolated liver nonparenchymal cells were washed in PBS buffer containing 10% FBS, and red cells were removed. The cells were stained with specific antibody: APC/cy7 anti-CD45 (1:400, BioLegend, 157204, Clone:30-F11), FITC anti-CD11b (1:400, BioLegend, 101205, Clone: M1/70), APC anti-F4/80 (1:400, eBioscience, 17-4801-82, Clone:BM8) at 4 °C for 30 min protected from light. For intracellular staining, cells were then fixed and permeabilized using the Foxp3 Fixation/Permeabilization kit (eBioscience). Intracellular antibodies were anti-HIF-2α PE (1:400, Novus, NB100-122PE, Polyclonal). Cells were then washed with cold PBS 3 times, and analysed by flow cytometry using FACS SORP flow cytometry (BD). The data were analysed using FlowJo software (TreeStar).

## Dual-luciferase reporter assay

Cells were seeded into a 48-well plate at a density of $2 \times 10^4$ per well. The luciferase constructs for the HIF response element (HRE), the oxygen-stable HIF-2α triple mutant (HIF-2α™) plasmid and the HIF-1α™ plasmid (constitutively active HIF-1α triple mutants) were previously described[12,26]. To explore the effect of So(d18:1) on the transcriptional regulatory activity of HIF-2α, HIF-2α™ plasmid, p2.1 HRE-Luc plasmid and Renilla positive control plasmid mixed with Lipo8000 transfection reagent were added to each well cells. To explore the effect of So(d18:1) on the transcriptional regulatory activity of HIF-1α, HIF-1α™ plasmid, p2.1 HRE-Luc plasmid and Renilla positive control plasmid mixed with Lipo8000 transfection reagent were added to each well cells.

For the Mammalian Two-Hybrid System, the pG5 luciferase vector was co-transfected with pBIND-HIF-2α and pACT-ARNT into cells using the protocol described in the CheckMate™ Mammalian Two-Hybrid System (Promega)[14].

For the mutant assay, HIF-2α™ plasmid and mHIF-2α S304M + G323E plasmid were used. For the *Cpt1a* rHRE binding assay, the pGL3 basic vector (Promega) was cloned with the presumed rHRE1 region in

the *Cpt1a* promoter upstream of the firefly luciferase gene as the reporter plasmid. The reporter plasmid, HIF-2α™ plasmid or corresponding control empty vector were transfected into HEK293T cells together. The luciferase assay was performed as previously described.

The cells were treated with a control vehicle, 2 μM HIF-2α-specific inhibitor PT2385, and 20 μM So(d18:1), So(d16:1), Sa(d18:1), So(d20:1) for 24 h, the supernatant was discarded, and the samples were gently rinsed with PBS buffer. Next, 100 μL of PLB lysis solution was added to the cells, and they were incubated at room temperature for 10 min. Ten microlitres of the cell lysate were added to a white flat-bottomed 96-well plate, and the following procedure was used in the multifunction microplate reader (Tecan): 40 μL of luciferase substrate was added, the fluorescence value was detected, and 40 μL of stop liquid was added. Finally, the ratio of the two fluorescence values was calculated.

## Mass spectrometry

Targeted lipidomics were performed according to a previous study with minor modifications[27]. Metabolic analysis of serum samples collected from NASH patients ($n = 16$) and healthy control ($n = 16$).Plasma (25 μL) or Liver tissue (20 mg) was added to 80 μL of water and homogenized for 1 min. Then, 400 μL of chloroform and methanol (v/v, 2:1) were added, and the samples were vortexed for 10 min and centrifuged at 4 °C and 12,000 rpm for 10 min. The lower layer was transferred into a new 1.5-ml tube and dried by a SpeedVac. Subsequently, 100 μL of cold methanol and isopropanol (v/v, 4:1) was added, and the tubes were vortexed for 10 min and centrifuged at 4 °C and 18,000 rpm for 10 min. The supernatant was transferred to a vial for MS detection. For plasma (100 μL), 400 μL of chloroform and methanol (v/v, 2:1) were added, and the remaining processes were the same as for liver tissue. A Waters UPLC BEH C18 column (2.1 mm (inner diameter) × 100 mm (length), 1.7 μm (particle dimension)) was used for separation. The mobile phase consisted of water (containing 5 mM ammonium acetate and 0.1% formic acid; phase A) and isopropanol:acetonitrile (1:1, v/v, containing 5 mM ammonium acetate and 0.1% formic acid; phase B) at a flow rate of 0.4 ml/min and a column temperature of 40 °C, with an injection volume of 2 μL. The UPLC and MS parameters used were chosen according to a previous study[27].

For the quantification of ceramides, S1P and sphingosine, 25 μL of plasma or 20 mg of liver tissue was homogenized with 400 μL of chloroform and methanol (v/v, 2:1) containing 5 μM sphingosine-d7 d18:1 and 25 μM ceramide-d7 d18:1/15:0 (Avanti Polar Lipids) as the internal standards. The mixture was oscillated immediately and then centrifuged at 13,000 rpm for 20 min. The lower phase was dried using a SpeedVac. The sediment was dissolved in 100 μL of isopropanol and acetonitrile (v/v, 1:1) and analysed using the Waters Acquity UPLC coupled with the AB SCIEX QTRAP 5500 system using a Waters UPLC CSH C18 column (3.5 μm, 2.1 × 100 mm). The UPLC and MS parameters used were chosen according to a previous study[28]. The lipid metabolites were quantified using MultiQuant 2.1 (AB SCIEX).

## NAS scoring

The NAS, also known as the NAFLD activity score (NAS), is calculated as the sum of three histological components, that is, steatosis (0–3), ballooning (0–2) and lobular inflammation (0–3). Patients with NAS ≥ 5 were considered definite NASH, patients with scores of 3 or 4 were considered borderline NASH, and patients with scores of less than 3 were diagnosed as NAFL.

## Enzyme-linked immunosorbent assay (ELISA)

The levels of IL-1β (Abclonal, RK00006) and IL-18 (Abclonal, RK00104) were measured by ELISA kits according to the manufacturer's instructions. In short, the standard or sample was added to the antibody-coated plate and incubated at 37 °C for 120 min. Bio-coupled antibody solution, avidin HRP solution and TMB substrate solution were added to the microporous plate in turn. The absorbance at

450 nm was measured within 15 min after adding the termination solution.

## LDH release detection

As for LDH release detection, we use 5 mM APAP and 20 μM So(d18:1) to treat the primary hepatocytes for 24 h. Collect the supernatant and use the LDH Cytotoxicity Assay Kit (Cat.C0016, Beyotime) to measure the cytotoxicity levels.

## Western blot and immunoprecipitation

Whole-cell lysates were prepared with RIPA buffer. The cell homogenate was incubated on ice in RIPA buffer for 15–20 min and then centrifuged at 10,000 rpm at 4 °C for 10 min. The supernatant was transferred into a new tube and mixed with 5× loading buffer. The mixture was boiled for 10 min.

For co-IP, Protein A/G PLUS agarose beads (Santa Cruz) were placed in the cell lysate supernatant. The samples were incubated upside down overnight at 4 °C. TBST buffer was used to wash 3 times. Then, 50 μL of 2× loading buffer was added to the beads and boiled for 10 min.

Each well containing 50 μg of protein lysate was separated by SDS–PAGE, transferred to a nitrocellulose membrane, and immunoblotted at 4 °C overnight. The antibodies were anti-caspase-1 (1:1000, CST, #24232, Monoclonal), anti-cleaved-caspase-1 (1:1000, CST, #89332, Monoclonal), anti-HIF-2α (1:1000, Novus, NB100-132, Clone: ep190b), anti-HIF-1α (1:1000, Proteintech, 20960-1-AP, Polyclonal), anti-ARNT (1:1000, Santa Cruz, sc-55526, Monoclonal), anti-GAPDH (1:1000, CST, #5174, Monoclonal) and anti-β-Actin (1:1000, Abclonal, AC038, Clone: ARC5115-01). The HRP-coupled secondary antibodies used were HRP-conjugated Goat anti-Mouse IgG (H+L) (1:2000, ABclonal, AS003), and HRP-conjugated Goat anti-Rabbit IgG (H+L) (1:2000, ABClonal, AS014) antibodies and immunoblotting was carried out using a chemical imaging system (ChemiDoc, Bio-Rad).

## RT-qPCR analysis

Liver tissues were flash-frozen in liquid nitrogen and stored at −80 °C. Total RNA from frozen liver tissues was extracted using TRIzol reagent (Invitrogen). cDNA was synthesized from 2 μg of total RNA using 5× All-In-One RT MasterMix (Abm). A list of quantitative PCR (qPCR) primer sequences is provided in Supplementary Table 2. The relative amount of each mRNA was compared to the corresponding gene and normalized, and the results are expressed as fold changes relative to the control group.

## RNA sequencing and analysis

Library preparation and transcriptome sequencing were conducted by GENEWIZ LLC. The Illumina HiSeq platform was used for sequencing. For the data analysis, we first evaluated the quality of the sequence data by fastqc v0.11.9, and the sequence quality was considered to be good for subsequent analysis. Trim-galore v0.6.7 was used for adaptor trimming and low-quality reads. Clean read mapping was conducted by Hisat2 v2.2.1, and we used mm10 as the mouse reference genome. After that, gene expression was quantified by featureCounts v2.0.1. All downstream analyses were performed in R v4.2.1. We used the edgeR v3.38.4 R package for differential expression analysis. We set the cutoff of differentially expressed genes as follows: *p*-value of 0.05 and absolute value of fold change of 1.5. Gene Ontology (GO) enrichment analysis and transcription factor enrichment analysis were conducted by the clusterProfiler v4.4.4 R package. We used the ARCHS4 transcription factor coexpression database from the Enrichr library as the database for transcription factor enrichment analysis. The record GSE262135 has been submitted to the GEO database.

## Single-cell RNA-sequencing analysis

We downloaded the count matrix of the GSE166504 dataset from the GEO database and analysed it using R. This is a single-cell

transcriptome dataset of livers where mice were fed a chow-diet, an HFHFD diet for 15 weeks, and an HFHFD diet for 30 weeks. We clustered these cells by mRNA expression level using the Seurat package, and then we annotated these cell clusters using the SingleR package.

## Statistics analysis

This study used GraphPad Prism software v.9.0. and SPSS software v.27.0 for analysis and statistics. The experimental results of this study are presented as the mean ± standard error of the mean (SEM). First, the Kolmogorov–Smirnov statistical method was used to detect the normality of all data. If the data conformed to a normal distribution, Student's *t*-test was used to compare two groups, one-way ANOVA was used for three or more groups, and Tukey's post-test was used for statistical analysis. If the data did not conform to a normal distribution, a nonparametric test was used. Mann–Whitney's statistical method was used for analysis between two groups, and the Kruskal–Wallis and Dunn's time tests were used for statistical analysis of three or more groups.

## Reporting summary

Further information on research design is available in the Nature Portfolio Reporting Summary linked to this article.

## Data availability

All of the data supporting the findings of this study are included in the Article and Supplementary information. The RNA-sequencing data generated in this study have been deposited in the GEO database under accession code GSE262135. The data generated in this study are provided in the Supplementary Information/Source Data file. The Single-cell RNA-sequencing data used in this study are available in the GEO database under accession code GSE166504. Source data are provided with this paper.

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

## Acknowledgements

This work was supported by the National Natural Science Foundation of China (No. 31925021, 82130022, 92357305 and 82341226 to C.J., 82288102, 82022028 and 82171627 to Y.P., 91857115, 81921001 and 92149306 to C.J.), and the National Key Research of Development Program of China (No. 2018YFA0800700 and 2022YFA0806400 to C.J.).

## Author contributions

Y.P. and J.X. conceptualized and designed the study. J.X, H.C., X.W., W.C., F.X., Q.N., C.Y., B.Z., M.Z., CY.Y., G.Z., Y.M., Y.W., X.Z., S.Y. and F.L. performed the experiments and analysed the data. Y.P. and H.R. supervised the study. H.C. and J.X. wrote the manuscript with input from all authors. J.L., XM.W., C.Z., P.X., C.J. and L.S. revised the manuscript. J.X, H.C. and X.W. contributed equally to this work. All authors edited the manuscript and approved the final manuscript.

## Competing interests

Y.M., Y.W. and X.Z. are employees of the Mengniu Institute of Nutrition Science. This study is supported in part by a research fund from the Mengniu Institute of Nutrition Science. The other authors declare no competing interests.
