## [Peer Review File · Nature Communications]

Sphingosine d18:1 Promotes Nonalcoholic Steatohepatitis by Inhibiting Macrophage HIF-2 αREVIEWER COMMENTS

Reviewer #1 (Remarks to the Author):

This first section of this manuscript describes the identification of a specific lipid (sphingosine [So]18:1) whose plasma levels are associated with NASH progression (in humans and mouse models). The second section is a series of mechanistic studies aimed at understanding how So18:1 promotes NASH progression. The conclusion is that So18:1 promotes NLRP3 activation in macrophages through affecting Hif2a levels/activity. There are many aspects of this data set that are interesting and that have the potential to be an important contribution to the field, but I think key additional control experiments are required to support the conclusions.

The data presented in Figure 1 and Supp Figure 1 on the identification of So18:1 is fairly compelling and sets up the subsequent mechanistic work well. However, the subsequent mechanistic studies fall short of proving the thesis of the manuscript. The major concern (across many of the studies) is the nature of the vehicle used when administering So18:1. It is not explicitly stated what this is, but in my opinion the authors should be using the other lipids they ID from the metabolomics, but that are not associated with NASH progression. Only by doing this can they be sure that the effects they see across many of the experiments (Figure 2, Figure 3, Figure 6) are due to specific effects of So18:1. Such evidence would provide a far more compelling argument that So18:1 is a specific driver of NASH progression.

The HIF2a KO and overexpression studies in Figure 3 and 4 are also quite convincing, but how this phenotype is related to So18:1 is not clear. For example, the authors do not provide evidence that the expression of Hif2a is decreased in macrophages from So18:1 treated mice or NASH mouse models. Finally, the suggestion that the So18:1 and Hif2a effects are driven through NLRP3 inflammasome activation are overstated, particularly in vivo.

My specific comments are also outlined below.

Line 32, sphingomyelins are the predominant sphingolipid, please add to your statement.

For the heat maps in Figure 1 it should be stated what the values are, are these z-scores, concentrations, something else, it is not known from the figure or legend. I would personally move the data in Sup Figure 1D, 1E and 1J into the main text, this is important and strong evidence.

Line 70. I think this is a one-sided interpretation. One could equally postulate that because the liver So18:1 levels decrease over the course of the CDAA-HFD, and that liver enzymes also increase over this time, perhaps the increased plasma levels are the result of hepatocyte damage and release into the circulation. I'm not saying this is the case, but a more balanced interpretation may be warranted.

There is a lack of description of how the sub-class of disease progression (e.g. the data in Supp 1B) data is obtained. Is this a re-classification of the NASH group, I assume so, but please be more descriptive in the text. Also, I would like to see the concentration data (i.e. that shown in Supp Figure 1D) for the other Sa and So species that you show in the heat map. They all look to change to a very similar extent. Also, I would re-format the data in Supp Figure 1B/C to look like Supp Figure 1E. Collectively, these changes will make it easier for the reader to see that while a number of sphingolipids changes in NASH, only So18:1 actually increases during disease progression.

Line 83. The conclusion is not convincing based on the data. Indeed, from a statistical point of view, they are not different. The conclusion should be tempered.

Line 88. 'accepted' is rather an odd phrasing. Perhaps just say mice 'were injected with'.

The data in Figure 2 seem reasonably robust, but I have a number of issues that I think should be addressed. (i) what is the nature of the vehicle? It does not say in the methods what this is; indeed, the methods only state that the So treated group were injected with So everyday, while there is no mention of the vehicle treated group. So, are these mice not being injected at all? If they are, what are they receiving? (ii) Relatedly, I'd very much like to see other lipids that the authors identified in the data in Figure 1 being used as controls – i.e. use lipids that are not associated with NASH progression. This would present a far stronger case that So18:1 has specific effects. (iii) You need to perform the metabolomics to show that your injections are increasing the plasma concentrations of So18:1, and also to what extent, they may be increased to levels far beyond what you see endogenously. Ideally, you want your So18:1 injections to increase plasma So18:1 within pathological levels. (iiii) What is happening to lipid levels in the liver following these injections? Collectively, I think several additional experiments are required to convincingly show that So18:1 specifically has the claimed effects.

Figure 3A,B. n numbers are very low, this is obviously from a single experiment. I would like to see a greater sample number used. Additionally, my comments above about the vehicle control apply here also. So, I would like to see the authors using the other So and Sa species they identify as being increased in NASH, but not increasing during disease progression, to show that the specific pro-inflammatory/immune recruitment effects they see are specifically due to So18:1.

Figure 3E-I. Same comment as above, repeat with other lipids to show the effect is specific to So18:1.

Line 149 seems to be the first mention of the GAN diet. Please define.

Line 148-149. You are not testing this. You are only determining if Hif2a contributes to NASH. You don't present any evidence that the deletion of Hif2a exacerbates NASH due to inflammasome activation. To support such a claim you should at least show that various inflammasome components, within macrophages, are increased in the Hif2a KO model – yes, il-1b is up a bit but so are lots of other inflammatory markers and this measure is just in a whole liver homogenate, not isolated cells. While inflammatory pathways are activated in Hif2a KO cells, figure 3E shows a large reduction in several markers of M2 macrophages, this is also likely to be a major reason for the potentiated inflammation in the Hif2a KO model – i.e. you are altering the balance of pro/anti-inflammatory macrophages. Furthermore, while you show that So18:1 decreases Hif2a in vitro, you do not show that either So18:1 administration in vivo or the CDAA-HFD diet reduced Hif2a in macrophages in vivo. Nor do you show that blocking NLRP3 inflammasome activity can reverse the effects described in Figure 4.

Line 221, I'm again concerned that a solvent is not the most appropriate control. Your whole thesis is that So18:1, and not other closely related lipids, promote NASH progression. Therefore, the best control would be the other related lipids. If you could show that So16:1 and some of the Sn lipids do not have the same effects as So18:1, this would present a far more compelling argument that So18:1 has specific properties not shared by other related lipid species.

Your central argument is that So18:1 decrease Hif2a protein levels (e.g. figure 2F). Yet looking at the input controls for your IP experiment in Figure 6C, it does not appear to do so. What is the reason for this apparent discrepancy?

Line 229. You somewhat establish this in vitro (see my previous comments about appropriate lipid controls), you do not provide evidence that (a) hif2a is reduced in vivo, either in response to So18:1 or in the NASH model, (b) you don't demonstrate that the effects of manipulating Hif2a expression (KO/over-expression) on NASH are via effects on NLRP3.

Reviewer #2 (Remarks to the Author):

In the current study, Xia et al. reported the association between plasma sphingosine d18:1 [So(d18:1)] levels and the severity of NASH. They also found that So(d18:1) aggravated the NASH phenotype in mice. Regarding the mechanism, they studied the effects of So(d18:1)-promoted activation of the NLRP3 inflammasome by inhibiting HIF-2a activity in macrophages. They found that So(d18:1) directly bound

with HIF-2a and inhibited the interaction of HIF-2a and ARNT. This study integrates clinical patient data and animal experiments. In general, this study provides a novel serological indicator for NASH diagnosis and novel targets for NASH treatment. However, there are several critical issues that should be addressed.

1. So(d18:1) increased ALT and AST levels in CADD-HFD-fed mice. Did So(d18:1) directly promote hepatocyte injury during NASH?
2. The possibility that So(d18:1) directly acts on other liver cell types should be discussed.
3. Did So(d18:1) treatment change ceramide and S1P levels in macrophage?
4. In Line 116, the author mentioned So(d18:1) promoted macrophage activation. To support this statement, did So(d18:1) upregulate inflammatory gene expression from the RNA-seq data?
5. Could macrophage-specific HIF-2a overexpression confront the effects of So(d18:1) on NASH in vivo? This is important because So(d18:1) may also act on other cell types, and this experiment could demonstrate the central role of macrophages in this process.
6. Regarding to “HIF-2 α plasmid with two proven missense mutations in the pocket which disabled other molecules to bind with HIF-2 α ”, the mutations need to be described in detail and the related literature should be cited.
7. The direct binding of So(d18:1) with HIF-2a is an important mechanism of this study. However, the authors only provide docking model to predict the direct binding, which is not enough. The evidence of direct binding of So(d18:1) with HIF-2a should be provided (such as SPR or ITC). It will be helpful if the disturbed binding of HIF-2a with mutated HIF-2a is also proven by experiments.
8. From the input of Figure 6C, So(d18:1) did not influence the protein level of HIF-2a. Was HIF-2a exogenous overexpressed in this experiment? If so, this should be clearly described in Figure legends or in method section.
9. Regarding Figure 6F, the figure shows that when HIF-2a was overexpressed, So(d18:1) increased luciferase activity. However, the authors describe the data as “similar to the fluorescence values of the empty plasmid” in line 251. In addition, in the same paragraph, there seems to be 4 groups of treatment in Figure 6F, however, the authors only showed 2 groups. Pleased check this result.

Minor

1. Figure legend of Fig. 2, “Il1b in J, statistical analysis was performed using two tailed Mann-Whitney U-tests”. Il1b is in panel K not J.
2. For the western blot of HIF-2a, why there are two bands in Figure 3F and one band in Figure 6C?

Reviewer #3 (Remarks to the Author):

In this study, Jialin Xia et al. found that sphingosine d18:1 (So(d18:1)) is elevated in the plasma of NASH patients as well as in the plasma of mice subjected to a NASH mouse model. Authors claim that this So(d18:1) elevation results in liver inflammation and fibrosis and that the pro-fibrotic and pro-inflammatory potential of sphingosine d18:1 rely on its ability to block HIF2a and HIFbeta heterodimerization and therefore HIF2 activity. Along this line authors clearly show that macrophage HIF2 inactivation leads to liver inflammation and fibrosis. These data are interesting but the following comments should be addressed.

1.- In Figure 3F, authors should run in parallel samples of BMDMs obtained from LysM-HIF2aLSL/LSL and HIF2aLysM mice in order to confirm that the two bands shown in the western blot correspond to HIF2a.

2.- Authors should show whether gene expression of Arginase 1, VEGF-a, Spint, Depdc7 and IL-10 are elevated in LysMHIF2aLSL/LSL BMDMs or reduced in LysMHIF2a deficient BMDMs.

3.- Authors show that So(d18:1) blunted HIF2a activity in BMDMs. But is this also in So(d18:1)-treated mice? Authors should try to perform liver immunostaining using antibodies against HIF2 (or HIF2 target genes) in costaining with macrophage markers in order to evaluate whether macrophage HIF2 expression/activity is really reduced in macrophages of So(d18:1)-treated mice in vivo. Or alternatively assess this point in macrophages isolated from the liver of control and So(d18:1)-treated mice. If these experiments reveals a partial decline of HIF2a, authors might assess whether macrophage-HIF2 heterozygous mice also results in increased liver inflammation and fibrosis.

4.- Authors should further assess whether So(d18:1) inhibits specifically HIF2 activity but not HIF1a. Along this line - in Figure 6C - authors should assess whether HIF1a and ARNT heterodimerization is affected by So(d18:1)?. Moreover, in Figure 6B, authors should assess the effect of So(d18:1) on the luciferase activity driven by a pBIND-HIF1a construct.

5.- In page 8, authors mentioned 'We administered So(d18:1) intraperitoneally to mice for 1 week, results showed that So(d18:1) increased the proportion of liver macrophages among all immune cells (Figure S3C, 3A-C)". However it is not clear whether Figure S3C refers to HFD-fed mice or So(d18:1)-treated mice. Authors should clarify this point.

6.- Authors use a GAN diet to assess liver inflammation and fibrosis in macrophages HIF2a-deficient mice. Why authors not use CDAA-HFD diet as in other experiments presented in this study?. Moreover in figure S3A, authors used HFHFD diet. Authors should clarify why they used these different diets.

7.- In Figure 3A, control group correspond to CDAA-HFD-fed mice?

Response to Reviewers' Comments

Reviewer #1 (Remarks to the Author):

This first section of this manuscript describes the identification of a specific lipid (sphingosine [So]18:1) whose plasma levels are associated with NASH progression (in humans and mouse models). The second section is a series of mechanistic studies aimed at understanding how So18:1 promotes NASH progression. The conclusion is that So18:1 promotes NLRP3 activation in macrophages through affecting Hif2a levels/activity. There are many aspects of this data set that are interesting and that have the potential to be an important contribution to the field, but I think key additional control experiments are required to support the conclusions.

Response: We appreciate the comments from reviewer #1 on our manuscript. We believe that all concerns raised by the reviewer can be addressed by additional rigorous experimentation and more detailed discussions. We have added the following experiments:

- 1) regarding the control for So18:1 administration, we added So(d16:1) as control to the animal experiment in Figures 2, and So(d16:1), So(d20:1) and So(d18:1) as control to the cell experiments in Figure 3 and Figure 6 to illustrate the specific effects of So(d18:1).
- 2) We used flow cytometry to detect the decrease of HIF-2a in liver macrophage in vivo under So(d18:1) treatment and in the NASH model. We have increased the sample size for flow cytometry analysis of liver macrophages in Figures 3A and 3B.
- 3) We conducted metabolomics studies to detect the plasma concentration of So(d18:1) in mice treated with vehicle and So(d18:1). Besides, we conducted lipidomics testing to determine what changes have occurred in the lipid levels in the liver after the treatment.

The data presented in Figure 1 and Supp Figure 1 on the identification of So18:1 is fairly compelling and sets up the subsequent mechanistic work well. However, the subsequent mechanistic studies fall short of proving the thesis of the manuscript. The major concern (across many of the studies) is the nature of the vehicle used when administering So18:1. It is not explicitly stated what this is, but in my opinion the authors should be using the other lipids they ID from the metabolomics, but that are not associated with NASH progression. Only by doing this can they be sure that the effects they see across many of the experiments (Figure 2, Figure 3, Figure 6) are due to specific effects of So18:1. Such evidence would provide a far more compelling argument that So18:1 is a specific driver of NASH progression.

Response:

We thank the reviewer believe that our data have the potential to be an important contribution to the field. In this original revision, we mainly used 0.5% CMC-Na

solution as solvent to make a suspension of So(d18:1) for the administration. We agree with the reviewer that using another lipid from the metabolomics as a control can better highlight the unique role of So(d18:1). We have found that So(d16:1), So(d20:1) and Sa(18:1) changed in the NASH patients' serum, but their concentrations were not increased as NASH progression (Fig1D, SupFig1C). So we used So(d16:1), So(d20:1) and Sa(d18:1) as control to the cell experiments and used So (d16:1) as control to the animal experiments in Figure 2, 3, and 6 to illustrate the specific effects of So(d18:1). We found only So18:1 promotes macrophages secreting inflammation factors through affecting HIF-2a levels/activity and promotes the progression of NASH, which reinforced the specific role of So(d18:1) in NASH progression.

The HIF2a KO and overexpression studies in Figure 3 and 4 are also quite convincing, but how this phenotype is related to So18:1 is not clear. For example, the authors do not provide evidence that the expression of Hif2a is decreased in macrophages from So18:1 treated mice or NASH mouse models. Finally, the suggestion that the So18:1 and Hif2a effects are driven through NLRP3 inflammasome activation are overstated, particularly in vivo.

Response:

We used flow cytometry to detect the expression of HIF-2a in macrophages in So(d18:1) treated mice or NASH mouse models. Data shows that the expression of Hif2a decreased in macrophages after So(d18:1) administration or NASH mouse models. The data and figures were added in the article now (Figure 3G and Supplementary Figure 3E). And as to NLRP3 activation, we agree that there will be other mechanisms that could affect NASH progression too. Thus, we revised the statement that the So(d18:1) and Hif2a effects are driven only by NLRP3 inflammasome activation in the article.

My specific comments are also outlined below.

1. Line 32, sphingomyelins are the predominant sphingolipid, please add to your statement.

For the heat maps in Figure 1 it should be stated what the values are, are these z-scores, concentrations, something else, it is not known from the figure or legend. I would personally move the data in Sup Figure 1D, 1E and 1J into the main text, this is important and strong evidence.

Response:

We've added Sphingomyelin as the main sphingolipids, and explained what the mean value of the heat map is in the figure legend of Figure 1. And we have moved the data Sup Figures 1D, 1E, and 1J into the main text in Figure 1E, 1F, 1G.

2. Line 70. I think this is a one-sided interpretation. One could equally postulate that because the liver So18:1 levels decrease over the course of the CDAA-HFD, and that

liver enzymes also increase over this time, perhaps the increased plasma levels are the result of hepatocyte damage and release into the circulation. I'm not saying this is the case, but a more balanced interpretation may be warranted.

Response:

We agree with the reviewer that the statement in line 70 is a one-sided interpretation. We changed the statement (now Line 121) and discussed the reason for the increase of sphingosine in plasma in discussion. In Supplementary figures, we tested the whole liver So(d18:1) content during the NASH model processing (SupFig 1C), and it showed a downward trend within the first two weeks, but no further change in the coming 6 weeks. We agreed that the increased plasma levels may be the result of hepatocyte damage and release into the circulation. Besides, according to existing articles, So(d18:1) is also present in feces (PMID: 32610095), so we infer that the increased So(d18:1) may originate from the intestine, too. These origins of So(d18:1) may exit together.

3. There is a lack of description of how the sub-class of disease progression (e.g. the data in Supp 1B) data is obtained. Is this a re-classification of the NASH group, I assume so, but please be more descriptive in the text. Also, I would like to see the concentration data (i.e. that shown in Supp Figure 1D) for the other Sa and So species that you show in the heat map. They all look to change to a very similar extent. Also, I would re-format the data in Supp Figure 1B/C to look like Supp Figure 1E. Collectively, these changes will make it easier for the reader to see that while a number of sphingolipids changes in NASH, only So18:1 actually increases during disease progression.

Response:

Yes, we used the NAS scores of the liver pathological slices to classify the disease progression. We added the description of the subgroup in the method (NAS scoring section) and the figure legend of Supplementary Figure 1. As for the heatmap, we used the intensity of response of each sphingolipids to make that figure. Although they all increased in NASH patients, So(d18:1) is the most significant one. And as for Supp Figure 1B/C (now supplementary Figure 1E-1I), we already redrawn them to match the format of Supp Figure 1E (now Figure 1F), indicating that only So(d18:1) truly increases during disease progression.

4. Line 83. The conclusion is not convincing based on the data. Indeed, from a statistical point of view, they are not different. The conclusion should be tempered.

Response:

We corrected and tempered the statement on line 83 (now line 132).

5. Line 88. 'accepted' is rather an odd phrasing. Perhaps just say mice 'were injected with'.

Response:

We have changed 'accepted' to 'injected'.

6. The data in Figure 2 seem reasonably robust, but I have a number of issues that I think should be addressed. (i) what is the nature of the vehicle? It does not say in the methods what this is; indeed, the methods only state that the So treated group were injected with So everyday, while there is no mention of the vehicle treated group. So, are these mice not being injected at all? If they are, what are they receiving? (ii) Relatedly, I'd very much like to see other lipids that the authors identified in the data in Figure 1 being used as controls – i.e. use lipids that are not associated with NASH progression. This would present a far stronger case that So18:1 has specific effects. (iii) You need to perform the metabolomics to show that your injections are increasing the plasma concentrations of So18:1, and also to what extent, they may be increased to levels far beyond what you see endogenously. Ideally, you want your So18:1 injections to increase plasma So18:1 within pathological levels. (iiii) What is happening to lipid levels in the liver following these injections? Collectively, I think several additional experiments are required to convincingly show that So18:1 specifically has the claimed effects.

Response:

(i) In this paper we mainly used 0.5% CMC-Na as the solvent. Control group was injected with the solvent every day. We have added this description in the method.

(ii) In Fig1D, we found that So(d16:1) increased in NASH patients serum, but they didn't show any future accumulation with the disease progression (SupFig 1H-I). In order to verify the unique function of So(d18:1), we chose So(d16:1) as a control lipid to administrate the NASH mice. As the figures showed in Fig2 and SupFig2, So(d16:1) injection didn't exacerbate NASH in mice.

(iii) We conducted metabolomics studies to detect whether So18:1 injection increases plasma concentration within the pathological range as expected. Data is showed in SupFig 2A.

(iiii) After injection of vehicles and So(d18:1), we conducted lipidomics testing to determine what changes have occurred in the lipid levels in the liver (Response Figure 1).

Response Figure 1: A, PLS-DA analysis of sphingolipids in liver of CDAA-HFD mice treated with vehicle or So(d18:1). B, VIP score plot of the difference sphingolipids between the two groups.

7. Figure 3A,B. n numbers are very low, this is obviously from a single experiment. I would like to see a greater sample number used. Additionally, my comments above about the vehicle control apply here also. So, I would like to see the authors using the other So and Sa species they identify as being increased in NASH, but not increasing during disease progression, to show that the specific pro-inflammatory/immune recruitment effects they see are specifically due to So18:1.

Response:

We have increased the sample size for flow cytometry analysis of liver macrophages in Figures 3A and B. And we supplied So(d16:1) as control in this experiment.

8. Figure 3E-I. Same comment as above, repeat with other lipids to show the effect is specific to So18:1.

Response:

We have supplemented other So and Sa species (So(d16:1), So(d20:1), Sa(d18:1)) in Figure 3E-I as controls (now figure 3E, supplementary figure 3G, 3H). As these So and Sa species (So(d16:1), So(d20:1), Sa(d18:1)) didn't influence the HIF-2a downstream genes and the protein level of IL-1 β and IL-18, we didn't detect their influence on HIF-2a and caspase-1 protein further.

9. Line 149 seems to be the first mention of the GAN diet. Please define.

Response:

GAN diet is Gubra Amylin NASH (GAN) diet, which is a high fat, high cholesterol, and high fructose feed (40% fat powered, 20% fructose, and 2% cholesterol), is fed to mice for more than 24 weeks (generally, steatosis begins at 16 weeks, NASH forms at 24 weeks, and fibrosis is induced at 26-32 weeks). We have added the definition of

GAN diet on Line 149 (now it is Line 213).

10. Line 148-149. You are not testing this. You are only determining if Hif2a contributes to NASH. You don't present any evidence that the deletion of Hif2a exacerbates NASH due to inflammasome activation. To support such a claim you should at least show that various inflammasome components, within macrophages, are increased in the Hif2a KO model – yes, il-1b is up a bit but so are lots of other inflammatory markers and this measure is just in a whole liver homogenate, not isolated cells. While inflammatory pathways are activated in Hif2a KO cells, figure 3E shows a large reduction in several markers of M2 macrophages, this is also likely to be a major reason for the potentiated inflammation in the Hif2a KO model – i.e. you are altering the balance of pro/anti-inflammatory macrophages. Furthermore, while you show that So18:1 decreases Hif2a in vitro, you do not show that either So18:1 administration in vivo or the CDAA-HFD diet reduced Hif2a in macrophages in vivo. Nor do you show that blocking NLRP3 inflammasome activity can reverse the effects described in Figure 4.

Response:

We changed the description of lines 148 to 149 (now line 212) as “To investigate whether HIF-2 α in macrophages can influence NASH disease progression”. Our previous work has demonstrated that the absence of HIF-2a leads to the activation of macrophage inflammasomes (PMID: 34433035), and many other reports have also found that the activation of NLRP3 inflammasomes is an important mechanism for the occurrence and development of NASH. We detected the levels of IL-1 β and IL-18 in macrophages of HIF2a $^{\Delta Lysm}$, which is consistent with our previous report (PMID: 34433035), indicating an increase in IL-1 β levels in macrophages of HIF2a KO. However, we also agree that NLRP3 inflammasome activation induced by reduced HIF-2a may be only one of the mechanisms by which HIF-2a deficiency leads to NASH. We would therefore modify the state of the conclusions and add discussions.

As to whether So(d18:1) also reduces HIF-2a in vivo, we added flow cytometry analysis of liver macrophages to verify (Fig 3G, SupFig 3E). Data shows that HIF-2a also reduced in liver macrophages.

Response Figure 2: A-B, Quantification of IL-1 β (A) and IL-18 (B) secretion from Hif2 $\alpha^{fl/fl}$ and Hif2 $\alpha^{\Delta Lysm}$ BMDMs that were transfected with two independent Nlrp3-

targeting siRNAs or a nontargeting siRNA after treatment with LPS and nigericin (n = 6). C, Representative immunoblot analysis of caspase-1 and IL-1 β from Hif2 $\alpha^{fl/fl}$ and Hif2 $\alpha^{\Delta Lysm}$ BMDMs that were stimulated with LPS and nigericin (n = 3).

11. Line 221, I'm again concerned that a solvent is not the most appropriate control. Your whole thesis is that So18:1, and not other closely related lipids, promote NASH progression. Therefore, the best control would be the other related lipids. If you could show that So16:1 and some of the Sn lipids do not have the same effects as So18:1, this would present a far more compelling argument that So18:1 has specific properties not shared by other related lipid species.

Response:

We have added other sphingolipids (So(d16:1), So(d20:1), Sa(d18:1)) as controls to repeat the HIF-2 α Luciferase assay for transcriptional activity test.

12. Your central argument is that So18:1 decrease Hif2a protein levels (e.g. figure 2F). Yet looking at the input controls for your IP experiment in Figure 6C, it does not appear to do so. What is the reason for this apparent discrepancy?

Response:

Because the Hif2a protein in Input (now Figure 6D) is exogenously overexpression. We transfected the HIF2a overexpression plasmid (oxygen stable HIF-2 α triple mutant (HIF-2 α^{TM}) plasmid) into the cells, so that there is stable and large amount of HIF2a present in the cell, which avoids the influence of HIF2a degradation on the experiment, and also amplifies the effect and makes the inhibition of HIF-2a binding with ARNT more obvious. However, in Figure 3F, HIF2a is endogenous and unstable. So(d18:1) inhibit the binding of HIF-2 α to ARNT, which impedes HIF-2 α entry into the nucleus for transcriptional regulation. HIF-2 α that remains in the cytoplasm is very easily hydrolysed and therefore protein levels are reduced.

13. Line 229. You somewhat establish this in vitro (see my previous comments about appropriate lipid controls), you do not provide evidence that (a) hif2a is reduced in vivo, either in response to So18:1 or in the NASH model, (b) you don't demonstrate that the effects of manipulating Hif2a expression (KO/over-expression) on NASH are via effects on NLRP3.

Response:

(a) We used flow cytometry to detect the decrease of hif2a in vivo, whether for So18:1 or in the NASH model (Fig3G, S3E). (b) Yes, we agree that NLRP3 may not be the only pathway which lead Hif2a KO to NASH. We have therefore modified the state about NLRP3 inflammation in the conclusion and discussion. We mainly focused on the impact of macrophage hif2a KO/overexpression on the progression on NASH in this paper.

Reviewer #2 (Remarks to the Author):

In the current study, Xia et al. reported the association between plasma sphingosine d18:1 [So(d18:1)] levels and the severity of NASH. They also found that So(d18:1) aggravated the NASH phenotype in mice. Regarding the mechanism, they studied the effects of So(d18:1)-promoted activation of the NLRP3 inflammasome by inhibiting HIF-2a activity in macrophages. They found that So(d18:1) directly bound with HIF-2a and inhibited the interaction of HIF-2a and ARNT. This study integrates clinical patient data and animal experiments. In general, this study provides a novel serological indicator for NASH diagnosis and novel targets for NASH treatment. However, there are several critical issues that should be addressed.

Response: We appreciate the comments from reviewer #2 on our manuscript. We believe that all concerns raised by the reviewer can be addressed by additional rigorous experimentation and more detailed discussions. We have added the following experiments: We used LysM-HIF-2a^{LSL/LSL} mouse experiments to test whether macrophage specific HIF-2a overexpression can counteract the effect of So(d18:1) on NASH in vivo. And we treated primary hepatocytes and LX-2 cell with So(d18:1) in vitro. Moreover, we conducted SPR experiments to verify the evidence of direct binding of So(d18:1) with HIF-2a. Besides, we have detected the level of ceramide and S1P levels after treatment of So(d18:1) on macrophages and analyzed the inflammatory gene expression from the RNA-seq data.

1. So(d18:1) increased ALT and AST levels in CADD-HFD-fed mice. Did So(d18:1) directly promote hepatocyte injury during NASH?

Response:

We used 20 μ M So(d18:1) to stimulate the primary hepatocytes directly and found that So(d18:1) didn't have a brutal ability to damage hepatocytes (SupFig 3I).

2. The possibility that So(d18:1) directly acts on other liver cell types should be discussed.

Response:

We used 20 μ M So(d18:1) to stimulate the primary hepatocytes and LX-2 cells, which are derived from human hepatic stellate cells, directly and found that So(d18:1) didn't have a brutal ability to damage hepatocytes (SupFig 3I), neither induced fibrosis in LX-2 cells (SupFig 3J). So(d18:1) may still have possibility directly on other liver cell types, such as endothelial cells, bile duct cells, T cells, B cells, dendritic cells, and so on. However, macrophage-specific HIF-2a overexpression confronted the effects of So(d18:1) largely (Figure 5), which could demonstrate the central role of macrophages in this process. We have added these into discussion.

3. Did So(d18:1) treatment change ceramide and S1P levels in macrophage?

Response:

So(d18:1) treatment didn't change the level of ceramide, while increased the concentration of S1P significantly in macrophages (Response Figure 3A-B). Besides, So(d18:1) treatment increased the concentration of So(d18:1) in macrophages as expected (Response Figure 3C). Furthermore, when we used a S1P synthesis inhibitor SKI178 to avoid the change of S1P totally, So(d18:1) treatment still inhibits transcriptional regulation function of HIF-2a significantly (Response Figure 3D). These results suggest that the function of treatment of So(d18:1) on macrophages mainly comes from So(d18:1) itself rather than the S1P or ceramide.

Response Figure 3: A-C, The concentration of total ceramide, S1P, So(d18:1) in macrophages after treatment of So(d18:1) at different time point. D, HRE-based luciferase assay in HEK293T transfected with HIF-2a, followed by treatment of PT-2385, So(d18:1) and So(d18:1) with S1P synthesis inhibitor SKI178.

4. In Line 116, the author mentioned So(d18:1) promoted macrophage activation. To support this statement, did So(d18:1) upregulate inflammatory gene expression from the RNA-seq data?

Response:

We analyzed the expression of inflammatory genes in RNA seq data and found that So(d18:1) didn't upregulate inflammatory gene expression of macrophages from the RNA-seq data. The result suggests that so (d18:1) didn't directly affect transcription of inflammation factors. BP pathway enrichment revealed So(d18:1) inhibits HIF-2 α -regulated signalling pathway (Figure 3C). So (d18:1) inhibits HIF2a downstream genes M2 macrophages marker genes, Arg1, Vegf, and so on, which may induce macrophages transition to an inflammatory state. On the other hand, So(d18:1) promotes synthesis and secretion of inflammatory factors but not transcription of inflammatory factors through HIF2a which promotes macrophage fatty acid oxidation, thereby promoting the second signal of inflammasomes (PMID: 34433035).

Response Figure 4: transcripts per million (Tpm) of inflammatory genes from the RNA-seq data.

5. Could macrophage-specific HIF-2a overexpression confront the effects of So(d18:1) on NASH in vivo? This is important because So(d18:1) may also act on other cell types, and this experiment could demonstrate the central role of macrophages in this process.

Response:

We used *LysMHif2a*^{LSL/LSL} mouse experiments to test whether macrophage specific HIF-2a overexpression can counteract the effect of So(d18:1) on NASH in vivo in Fig5 and SupFig5. Specifically, the mice were divided into four groups: *Hif2a*^{+/+}, *Hif2a*^{+/+} + So(d18:1), *LysMHif2a*^{LSL/LSL} and *LysMHif2a*^{LSL/LSL}+So(d18:1), all fed with CDAA-HFD. The level of ALT, AST, the lobular inflammation score, the fibrosis area of HE and sirus staining and the inflammation and fibrosis genes, all showed that macrophage-specific HIF-2a overexpression confronted the effects of So(d18:1) on NASH in vivo.

6. Regarding to “HIF-2 α plasmid with two proven missense mutations in the pocket which disabled other molecules to bind with HIF-2 α ”, the mutations need to be described in detail and the related literature should be cited.

Response:

We cited relevant literature in these part. The Missense mutations are S304M and G323E, which disabled the binding of other molecules, such as 1,3-diaminopropane and PT2385, with HIF-2a (PMID: 31708445; PMID: 27595394; PMID: 27595393).

7. The direct binding of So(d18:1) with HIF-2a is an important mechanism of this study. However, the authors only provide docking model to predict the direct binding, which is not enough. The evidence of direct binding of So(d18:1) with HIF-2a should be provided (such as SPR or ITC). It will be helpful if the disturbed binding of HIF-2a with mutated HIF-2a is also proven by experiments.

Response:

We conducted SPR experiments to verify the binding of So (d18:1) with HIF-2a. The result showed that So(d18:1) had a direct binding with HIF-2a and K_D (binding affinity) values were determined as 19.51 μM (Response Figure 5A). We also conduct SPR experimen to detect the binding of So(d18:1) with mutated HIF-2a (Response Figure 5B). The result showed that So(d18:1) had no binding with mutated HIF-2a which indicated that the S304M and G323E mutation of HIF-2a disturbed the binding of So(d18:1) with HIF-2a.

Response Figure 5: A, Surface plasmon resonance (SPR) indicated binding of So(d18:1) to the HIF2a. B, Surface plasmon resonance (SPR) indicated no binding of So(d18:1) to the mutant HIF2a.

8. From the input of Figure 6C, So(d18:1) did not influence the protein level of HIF-2a. Was HIF-2a exogenous overexpressed in this experiment? If so, this should be clearly described in Figure legends or in method section.

Response:

We clearly described that HIF-2a is exogenous overexpression in the Figure legends of Figure 6C (now Figure 6D).

9. Regarding Figure 6F, the figure shows that when HIF-2a was overexpressed, So(d18:1) increased luciferase activity. However, the authors describe the data as "similar to the fluorescence values of the empty plasmid" in line 251. In addition, in the same paragraph, there seems to be 4 groups of treatment in Figure 6F, however, the authors only showed 2 groups. Pleased check this result.

Response: In Figure 6F (now Figure 6G), The Y axis of this figure represents the ratio

of fluorescence values between overexpressing plasmids to empty plasmids. In the control group, due to the inhibition of *cpt1a* by overexpressed HIF-2 α , the fluorescence value of HIF-2 α overexpression plasmid group was lower than that of the empty plasmid group, so their ratio was close to 0.5; When administered with So(d18:1), the effect of overexpressed HIF-2 α was inhibited by So(d18:1). Therefore, the fluorescence value of the overexpressed plasmid group is approximately equal to the fluorescence of the empty plasmid group, so their ratio is close to 1. As the Y axis of this figure represents the ratio of two groups, so the figure only showed 2 groups rather than 4 groups.

Minor

1. Figure legend of Fig. 2, “II1b in J, statistical analysis was performed using two tailed Mann-Whitney U-tests”. II1b is in panel K not J.

Response:

We have changed J to K.

2. For the western blot of HIF-2 α , why there are two bands in Figure 3F and one band in Figure 6C?

Response:

Figure 3F shows endogenous HIF-2 α protein, which is more delicate and can be influenced by post-translational modifications, post-translational cleavage and relative charge. Figure 6C (now Figure 6D) shows exogenous HIF-2 α protein derived from the overexpression plasmid, oxygen-stable HIF-2 α triple mutant (HIF-2 α^{TM}) plasmid, which is more stable and hard to be degrade.

Reviewer #3 (Remarks to the Author):

In this study, Jialin Xia et al. found that sphingosine d18:1 (So(d18:1)) is elevated in the plasma of NASH patients as well as in the plasma of mice subjected to a NASH mouse model. Authors claim that this So(d18:1) elevation results in liver inflammation and fibrosis and that the pro-fibrotic and pro-inflammatory potential of sphingosine d18:1 rely on its ability to block HIF2 α and HIFbeta heterodimerization and therefore HIF2 activity. Along this line authors clearly show that macrophage HIF2 inactivation leads to liver inflammation and fibrosis. These data are interesting but the following comments should be addressed.

Response: We appreciate the comments from reviewer #3 on our manuscript. We believe that all concerns raised by the reviewer can be addressed by additional rigorous experimentation and more detailed discussions. We have added the following experiments: We run in parallel samples of BMDMs obtained from *LysMHIF2a^{LSL/LSL}* and *HIF2a ^{Δ LysM}* mice to confirm that the two bands shown in the western blot correspond to HIF2 α . And we detected gene expression of Arginase 1, VEGF-a, Spint,

Depdc7 and IL-10 in *LysMHif2a^{LSL/LSL}* BMDMs. Moreover, we performed flow cytometry to detect the HIF2a levels in liver macrophages. Besides, we used luciferase assay to illustrate that So(d18:1) inhibits specifically HIF2a activity but not HIF1a. We evaluated whether HIF1a and ARNT heterodimerization are affected by So(d18:1) by co-IP and pBIND-HIF1a construct.

1.- In Figure 3F, authors should run in parallel samples of BMDMs obtained from *LysM-HIF2a^{LSL/LSL}* and *HIF2a^{LysM}* mice in order to confirm that the two bands shown in the western blot correspond to HIF2a.

Response: We run in parallel samples of BMDMs obtained from *LysMHIF2a^{LSL/LSL}* and *HIF2a^{ΔLysM}* mice to confirm that the two bands shown in the western blot correspond to HIF2a.

Response Figure 6: representative immunoblot analysis of HIF-2a of BMDMs from stimulated with vehicle and So(d18:1) isolated from wild-type mice and BMDMs isolated from *LysMHif2a^{LSL/LSL}* and *Hif2a^{ΔLysM}* BMDMs.

2.- Authors should show whether gene expression of Arginase 1, VEGF-a, Spint, Depdc7 and IL-10 are elevated in *LysMHIF2a^{LSL/LSL}* BMDMs or reduced in *LysMHIF2a* deficient BMDMs.

Response:

The gene expression of Arginase 1, VEGF-a, Spint, Depdc7 and IL-10 were elevated in *LysMHif2a^{LSL/LSL}* BMDMs. They all increased because of *Hif2a* overexpression.

Response Figure 7: Relative mRNA levels of *Hif2a* and its downstream target genes in *Hif2a^{+/+}* or *LysMHif2a^{LSL/LSL}* BMDMs.

3.- Authors show that So(d18:1) blunted HIF2a activity in BMDMs. But is this also in

So(d18:1)-treated mice? Authors should try to perform liver immunostaining using antibodies against HIF2 (or HIF2 target genes) in costaining with macrophage markers in order to evaluate whether macrophage HIF2 expression/activity is really reduced in macrophages of So(d18:1)-treated mice in vivo. Or alternatively assess this point in macrophages isolated from the liver of control and So(d18:1)-treated mice. If these experiments reveals a partial decline of HIF2a, authors might assess whether macrophage-HIF2 heterozygous mice also results in increased liver inflammation and fibrosis.

Response:

We performed flow cytometry to detect the HIF2a levels in liver macrophages (Fig 3G and SupFig 3E). These suggested that macrophage HIF2 expression/activity is really reduced in macrophages of So(d18:1)-treated or CDAA-HFD fed mice in vivo. As HIF2a^{ΔLysM} mice have a partial decline of HIF2a in BMDMs (Response Figure 6), so we used the HIF2a^{ΔLysM} mice to detected the liver inflammation and fibrosis (Figure 4).

4.- Authors should further assess whether So(d18:1) inhibits specifically HIF2 activity but not HIF1a. Along this line - in Figure 6C - authors should assess whether HIF1a and ARNT heterodimerization is affected by So(d18:1)?. Moreover, in Figure 6B, authors should assess the effect of So(d18:1) on the luciferase activity driven by a pBIND-HIF1a construct.

Response:

We used luciferase assay to illustrate that So(d18:1) inhibits specifically HIF2a activity but not HIF1a (Fig 6B). We evaluated whether HIF1a and ARNT heterodimerization are affected by So(d18:1) by co-IP (Fig 6E) and pBIND-HIF1a construct (Response Figure 8).

Response Figure 8: Schematic and experimental representation of TAY-cyclo or So(d18:1) mediated HIF-1a-ARNT interaction by pG5GAL4 luciferase assay followed by pBINDHIF-1a and pACT-ARNT transfection of HEK293T cells (n = 6).

5.- In page 8, authors mentioned ‘We administered So(d18:1) intraperitoneally to mice for 1 week, results showed that So(d18:1) increased the proportion of liver macrophages among all immune cells (Figure S3C, 3A-C)’. However it is not clear whether Figure S3C refers to HFD-fed mice or So(d18:1)-treated mice. Authors should clarify this

point.

Response:

Figure S3C is the gating strategy of liver macrophages from chow diet mice treated with vehicle. Figure 3A-B showed the flow cytometric and statistical analysis of chow-diet fed mice treated with control, So(d16:1) and So(d18:1). We have clarified this point in the Figure legend of Figure S3C and Figure 3A-B, as well as in the text of Line 169-172 in page 9.

6.- Authors use a GAN diet to assess liver inflammation and fibrosis in macrophages HIF2a-deficient mice. Why authors not use CDAA-HFD diet as in other experiments presented in this study?. Moreover in figure S3A, authors used HFHFD diet. Authors should clarify why they used these different diets.

Response:

CDAA-HFD food induced NASH inflammation and fibrosis are more severe. Since we found in Fig1 that So (d18:1) may be more closely related to inflammation and fibrosis compared to fat accumulation, we chose CDAA-HFD to highlight the phenotypic changes of inflammation and fibrosis. GAN diet has a longer processing time and is more in line with the disease development process of NASH in humans. Therefore, we chose this model to observe whether the impact of HIF-2a on NASH disease progression only affects inflammation and fibrosis, without affecting fat accumulation. But now, to further emphasise that macrophage-specific HIF-2a overexpression can resist NASH, we switched to a more intense CDAA-HFD.

7.- In Figure 3A, control group correspond to CDAA-HFD-fed mice?

Response:

In Figure 3A, the control group correspond to chow-diet fed mice. We have added the descriptions in the figure legend of Figure 3A.

REVIEWER COMMENTS

Reviewer #1 (Remarks to the Author):

The authors have done an excellent job of addressing my comments. They have added in the essential controls that I felt were necessary to more fully understand the specificity of So(18:1). The new data clearly show that indeed So18:1 has specific effects not shared by related lipids, thus supporting the main conclusion of the manuscript. They have also added essentially all the new pieces of data/experiments I suggested and made various textual changes, all of which strengthens the manuscript.

There remain a few odd phrases in the manuscript, for example line 407 describing the potential of So18:1 to causes cell death directly – “brutal ability”. The authors may wish to carefully check through the manuscript for other textual/grammatical issues. In the main the language is fine, just a few minor things could be corrected.

Reviewer #2 (Remarks to the Author):

The authors have addressed my concerns.

Reviewer #3 (Remarks to the Author):

Authors have addressed most of my comments. However, regarding my original comment #3, authors should demonstrate more convincingly that So(d18:1) reduces HIF2a expression in macrophages in vivo. Moreover, I am including a minor comment related to my original comments #1.

Regarding my original comment #1.

In the figure legend of the figure shown in the response letter, BMDMs isolated from LysMHif2 α LSL/LSL and Hif2 α Δ LysM were treated with vehicle?, is this correct? If so, this should be included in the figure legend. Moreover, this figure might be included in the Supplementary information section.

Regarding my original comment #3.

In Figure 3G and S3E, authors have assessed HIF2a expression by flow cytometry in macrophages isolated from So(d18:1)-treated or CDAA-HFD fed mice in vivo.

However, authors should include - as positive and negative control - BMDMs isolated from LysMHif2 α LSL/LSL and Hif2 α Δ LysM mice to confirm that signal detected by flow cytometry really corresponds to mouse HIF2a.

Alternatively, authors could use western blot instead of flow cytometry.

On the other hand, Figure legend 3G should be corrected to 'Flow cytometric and statistical analysis of HIF-2a in liver macrophages.....' instead of 'Flow cytometric and statistical analysis of HIF-1a in liver macrophages.....'

Finally, in the Figure legend S3E, 'chow dier' should be corrected to 'chow diet'.

REVIEWER COMMENTS

Reviewer #1 (Remarks to the Author):

The authors have done an excellent job of addressing my comments. They have added in the essential controls that I felt were necessary to more fully understand the specificity of So(18:1). The new data clearly show that indeed So18:1 has specific effects not shared by related lipids, thus supporting the main conclusion of the manuscript. They have also added essentially all the new pieces of data/experiments I suggested and made various textual changes, all of which strengthens the manuscript.

There remain a few odd phrases in the manuscript, for example line 407 describing the potential of So18:1 to causes cell death directly – “brutal ability”. The authors may wish to carefully check through the manuscript for other textual/grammatical issues. In the main the language is fine, just a few minor things could be corrected.

Response:

We have changed the “brutal ability” to “didn’t directly promote hepatocyte death”. We have carefully check through the manuscript for other textual/grammatical issues and corrected a few minor things.

Reviewer #2 (Remarks to the Author):

The authors have addressed my concerns.

Reviewer #3 (Remarks to the Author):

Authors have addressed most of my comments. However, regarding my original comment #3, authors should demonstrate more convincingly that So(d18:1) reduces HIF2a expression in macrophages in vivo. Moreover, I am including a minor comment related to my original comments #1.

Regarding my original comment #1.

In the figure legend of the figure shown in the response letter, BMDMs isolated from LysMHif2 α LSL/LSL and Hif2 α Δ LysM were treated with vehicle?, is this correct? If so, this should be included in the figure legend. Moreover, this figure might be included in the Supplementary information section.

Response:

Yes, BMDMs isolated from LysMHif2 α ^{LSL/LSL} and Hif2 α ^{Δ LysM} were treated with vehicle. We have included this in the figure legend of . And we have included the figure in the Supplementary information section.

Regarding my original comment #3.

In Figure 3G and S3E, authors have assessed HIF2a expression by flow cytometry in macrophages isolated from So(d18:1)-treated or CDAA-HFD fed mice in vivo. However, authors should include - as positive and negative control - BMDMs isolated from *LysMHif2a*^{LSL/LSL} and *Hif2a*^{ΔLysM} mice to confirm that signal detected by flow cytometry really corresponds to mouse HIF2a.

Alternatively, authors could use western blot instead of flow cytometry.

Response:

We have included BMDMs isolated from *LysMHif2a*^{LSL/LSL} and *Hif2a*^{ΔLysM} mice as positive and negative control to confirm that signal detected by flow cytometry really corresponds to mouse HIF2a.

On the other hand, Figure legend 3G should be corrected to 'Flow cytometric and statistical analysis of HIF-2a in liver macrophages.....' instead of 'Flow cytometric and statistical analysis of HIF-1a in liver macrophages.....'

Response:

We have corrected the Figure legend 3G as "Flow cytometric and statistical analysis of HIF-2a in liver macrophages".

Finally, in the Figure legend S3E, 'chow dier' should be corrected to 'chow diet'.

Response:

We have corrected the Figure legend S3E 'chow dier' to 'chow diet'.

REVIEWERS' COMMENTS

Reviewer #3 (Remarks to the Author):

The authors have satisfactorily addressed my final concerns.